# Incorruptible Neural Networks: Training Models that can Generalize to Large Internal Perturbations

## Abstract

Flat regions of the neural network loss landscape have long been hypothesized to correlate with better generalization properties. A closely related but distinct problem is training models that are robust to internal perturbations to their weights, which may be an important need for future low-power hardware platforms. Several methods have been proposed to guide optimization toward improved generalization, such as sharpness-aware minimization (SAM) and random-weight perturbation (RWP), which rely on either adversarial or random perturbations, respectively. In this paper, we explore how to adapt these approaches to find minima robust to a wide variety of random corruptions to weights. First, we evaluate SAM/RWP across a wide variety of noise settings, and in doing so establish that over-regularization during training is key to finding optimally-robust minima. At the same time, we also observe that large perturbations lead to a vanishing gradient effect caused by unevenness in the loss landscape, an effect particularly pronounced in SAM. Quantifying this effect, we map out a general performance trend of SAM and RWP, determining that SAM works best for robustness to small perturbations, whereas RWP works best for large perturbations. Lastly, to overcome the deleterious vanishing gradient effect during training, we propose a dynamic perturbation schedule which matches the natural evolution of the loss landscape and produces minima more noise-robust than otherwise possible.

## 1 Introduction

Optimizing deep neural network models in order to locate flat minima in the loss landscape has long been a problem of interest to researchers, driven by the theory that flat minima correlate with better generalization on unseen data Nitish Shirish Keskar & Tang (2016); Dziugaite & Roy (2017); Jiang et al. (2019). From this line of thought have emerged several modified methods for optimization that are explicitly designed to locate flat minima. The most prominent of these, sharpness-aware minimization (SAM), uses a one-step adversarial (i.e. *worst-case*) perturbation along the direction of the gradient to ensure low loss over a finite volume of the loss landscape, and has been demonstrated to successfully improve generalization in a wide array of settings Foret et al. (2021). A lesser-known alternative, known as random-weight perturbation (RWP), has likewise been studied as a potential mechanism for finding flat minima, but in contrast to SAM, has failed to show consistent generalization benefit Bisla et al. (2022); Li et al. (2024a).

While flat minima are primarily studied to improve generalization, they are also relevant to finding minima that are robust to *weight-space* perturbations. Although high-fidelity digital hardware has minimized this issue, the growing demand for compute has driven interest in analog neural network accelerators as energy-efficient alternatives to digital processors Sebastian et al. (2020). Of particular concern to analog in-memory computing (AIMC) solutions are irreducible analog hardware errors, which induce accuracy-degrading effects. These errors arise from many sources, such as quantization errors, memory-cell programming errors, and conductance drift- of particular focus to this paper are programming errors, which manifest as random perturbations to the model's parameters. Previous analog computing literature has identified this problem and proposed solutions for specific hardware configurations, relying on empirically-measured device error profiles to induce robustness for an exact deployment scenario Gokmen et al. (2019); Rasch et al. (2023). However, these works fail to

experimentally investigate a key idea underpinning their approach to noise-aware training: namely that the best test-time performance arises from faithfully emulating the expected test-time noise distribution during training (i.e. RWP using a matched noise distributions). We pose two natural questions in response: firstly, is the expected test-time noise distribution necessarily ideal when applied randomly during training, or can this distribution be more cleverly engineered? Secondly, we question whether or not random perturbations, in any form, are ideal for finding noise-robust minima, or if a more deliberate perturbation (as in the case of SAM) is in fact superior?

Naively, one might assume that the same methods for finding flat minima should remain applicable to the problem of weight perturbations. However, several subtle differences require us to consider the problem with extra care. First, generalizability is associated with flatness in the *training* loss landscape; in contrast, weight-noise robustness directly requires flatness in the *test* loss landscape. Second, dependent on the size of applied perturbations, a minimum may require a *significantly* greater degree of flatness over a larger volume than is needed for good generalization. As a result, these methods need to be used well outside the operating point that is typically studied. Lastly, the nature of the flat minimum should respect a known distribution of perturbations (i.e. the minimum should be flat where perturbations are most likely to occur). As such, the ability to produce a general robustness across a multitude of perturbation scenarios is another question of significant relevance.

In this paper, we investigate three key questions: first, is resilience to weight-space noise best addressed through exactly matching the form of training and test perturbations experienced? Second, can we infer an optimal training protocol based on the expected noise characteristics of inference? Lastly, can we understand the mechanisms through which these perturbative training methods achieve noise-robustness, and can we engineer improvements? Our initial experiments show that over-regularization, i.e. using training perturbations larger than test-time noise, yields optimally robust models, in contrast to the conventional wisdom on noise-robust training. Next, when comparing RWP with SAM, we find that SAM actually produces more noise-robust models for test-time noise below a certain strength threshold, even as it finds strictly sharper training minima. However, SAM is also more sensitive to the scaling of its perturbation, resulting in less-optimal minima under high test-time noise. We identify that increasing training perturbation strength causes a vanishing-gradient effect due to the rapidly increasing loss, with SAM being more affected due to its sharper perturbation direction. To mitigate this, we propose dynamic perturbation schedules for SAM and RWP, starting with small perturbations early in training to better align with the loss landscape.

In summary, our key contributions are as follows:

- We empirically show that optimal noise-robustness is not achieved through matching training/test perturbations, but rather through applying stronger perturbations (i.e. over-regularization) during training.

- We chart the relative benefit of SAM and RWP based on the strength of the test noise, showing that SAM is superior in weaker noise settings but poor in strong-noise settings.

- We identify a vanishing-gradient effect which arises from strong perturbations, degrading training particularly for SAM.

- We demonstrate that perturbative training can be further enhanced through an appropriately-selected dynamic perturbation schedule which increases as training proceeds produces significantly more noise-robust models.

- We validate our results using simulations of analog hardware, demonstrating improvement over the conventional approach in this domain.

## 2 RELATED WORK

### 2.1 CORRUPTION-ROBUST NEURAL NETWORKS

Many previous works have studied the robustness of neural networks to input corruptions, either adversarial Goodfellow et al. (2015); Mustafa et al. (2019); Yan et al. (2018) or random (common corruptions) Rusak et al. (2020); Fang et al. (2023); Kar et al. (2022); Mintun et al. (2021); Guo et al. (2023); Modas et al. (2022). However, few works have previously considered the setting of random noise corruptions to model weights. In the realm of AIMC, several prior works have examined

the resiliency of neural network inference accuracy to various sources of device noise Yang et al. (2022); Gokmen et al. (2019); Kariyappa et al. (2021); Rasch et al. (2023); Xiao et al. (2022b). The most comprehensive of these, Rasch et. al., applies regularization through injecting hardware noise emulating the target deployment platform during training. However, the methods these works propose are either application-specific or focus on hardware mitigations, and neglect to connect these ideas to the broader problem of finding flat minima.

## 2.2 Sharpness-Aware Minimization

SAM has recently emerged as a popular approach for finding flat, well-generalizing minima in the training landscape Foret et al. (2021). In contrast to standard SGD-based optimization methods, SAM first perturbs the model using a gradient ascent step, approximating a local maximization, followed by a descent step from the perturbed point. Since its inception, an array of modified variants aiming to improve SAM's generalization ability Liu et al. (2022b); Li et al. (2024b); Kwon et al. (2021); Kim et al. (2022) or training efficiency Du et al. (2022); Jiang et al. (2023); Liu et al. (2022a); Mi et al. (2022); Xie et al. (2024) have been proposed.

Several works have closely examined SAM's properties to better understand its empirically observed success. Andriushchenko et. al. studies the convergence of SAM through theoretically demonstrating the implicit biases of SAM when applied to a diagonal linear network Andriushchenko & Flammarion (2022). Wen et. al. more precisely demonstrates the measure of sharpness that the mini-batch variation of SAM regularizes Wen et al. (2023). Khanh et. al. provide a more comprehensive analysis of the convergence properties of SAM Khanh et al. (2024). Baek et. al. studies SAM's ability to induce robustness to label noise, demonstrating a positive effect from regularization of the network Jacobian Baek et al. (2024).

## 2.3 Random Weight Perturbation

Randomly perturbing weights during optimization has long been known as a straightforward approach to regularizing neural networks during training An (1996). Neelankantan et. al. demonstrate that adding noise to gradients during training improves generalization, achieving a similar effect to residual connections in deep networks Neelakantan et al. (2015). Zhou et. al. empirically show that weight noise can guide the optimization route out from spurious local minima Zhou et al. (2019). Bisla et. al. propose low-pass filtering the loss function (practically implemented through sampling weight noise from a Gaussian distribution) as a means to smoothing the loss landscape and hence improve generalization Bisla et al. (2022). Li et. al. propose a modified form of RWP which introduces filter-wise perturbations based on the historical graident Li et al. (2024a). Möllenhoff et. al. establish a theoretical connection between SAM and RWP in which SAM can be recovered through an optimal relaxation of the randomly perturbed objective Möllenhoff & Khan (2023).

## 3 Preliminaries

Consider a training dataset $\mathcal{S} = \{(\mathbf{x_i}, \mathbf{y_i})\}_{i=1}^n$ drawn i.i.d from a data distribution $\mathcal{D}$. Let $f(\mathbf{x}, \mathbf{w})$ be a neural network model with trainable parameters $\mathbf{w} \in \mathbb{R}^d$. Given some loss function on individual samples $l(f(\mathbf{x_i}, \mathbf{w}), \mathbf{y_i}) \in \mathbb{R}^+$, we define the empirical loss on the training dataset to be $L_{\mathcal{S}}(\mathbf{w}) = \frac{1}{n} \sum_{i=1}^n l(f(\mathbf{x_i}, \mathbf{w}), \mathbf{y_i})$. Our goal is to train a model $f$ which minimizes the *perturbed* distribution loss $L_{\mathcal{D}}^*(\mathbf{w}) = \mathbb{E}_{(\mathbf{x}, \mathbf{y}) \sim \mathcal{D}, \mathbf{p} \sim \mathbb{Q}}[l(f(\mathbf{x}, \mathbf{w} + \mathbf{p}), \mathbf{y})]$, where $\mathbb{Q}$ is a known distribution of possible weight perturbations. Although in principle $\mathbb{Q}$ can be arbitrary, for simplicity's sake we restrict $\mathbb{Q}$ to be a zero-mean isotropic Gaussian scaled by the maximum magnitude element in a given weight filter $w$, defined as $\mathcal{N}(0, \max_j |w_j| \sigma_q^2 \mathbb{I})$ with variance $\sigma_q^2$ (hereafter referred to as $\sigma_{test}$). We argue that the zero-mean Gaussian assumption faithfully captures the dynamics of analog programming errors: if the analog error distribution has a non-zero mean, it can be compensated for with a constant shift in the weights, whereas even if the error distribution is non-Gaussian, the sum is nonetheless Gaussian by means of the Central Limit Theorem. We directly scale the noise variance by the weight magnitude to more closely emulate inference on analog in-memory computing processors, in which the digital weights are mapped to a finite conductance range determined by the limitations of the hardware. As a result, the noise applied to the weights is fixed to a given fraction of the weight magnitude, and cannot be mitigated through simple re-scaling. We note that the perturbed

distribution loss *directly* reflects the flatness of the minima in parameter-space, hence the interest in applying both SAM and RWP.

**SAM** The modified SAM training loss can be expressed as follows:

$$L_{\mathcal{S}}^{SAM}(\mathbf{w}) = \max_{||\epsilon||_2^2 \le \rho} L_{\mathcal{S}}(\mathbf{w} + \epsilon) \tag{1}$$

where $\rho$ is a scalar hyperparameter corresponding to the radius of the $l_2$ ball in which the maximization is performed. In practice, the inner maximization within the SAM objective is not tractable, and instead approximated using a single gradient ascent step of length $\rho$.

**RWP** The RWP training loss we adopt in this paper is described by the following:

$$L_{\mathcal{S}}^{RWP}(\mathbf{w}) = \mathbb{E}_{\epsilon \sim \mathcal{N}(0, \max_j |w_j| \sigma_p^2 \mathbb{I})}[L_{\mathcal{S}}(\mathbf{w} + \epsilon)] \tag{2}$$

where $\sigma_p^2$ is the variance of the perturbation distribution (hereafter referred to as $\sigma_{train}$), left to our choosing. In our implementation, we sample one $\epsilon$ per minibatch, noting that the expectation is taken implicitly over the stochastic minibatches.

**Sharpness Measures** Given our problem setting, we use a simple measure for quantifying sharpness: the difference in the loss value between the original point on the optimization path and a given perturbed point. We use two varieties of this measure (originally defined in Wen et al. (2023)): *ascent-direction* sharpness (Eq. 3), in which the perturbed point lies along the gradient, and *average-direction* sharpness (Eq. 4), in which the perturbed point is sampled randomly, according to our noise distribution of interest. We also define *gradient sharpness* to refer to the sharpness at the point of perturbation (where the update gradient is computed) for SAM/RWP. In the case of SAM, its gradient sharpness is equivalent to ascent-direction sharpness, whereas for RWP it is equivalent to average-direction sharpness. For the sake of practicality, we use the approximate measure of $m$-sharpness, in which the sharpness is calculated through averaging over mini-batches of size $m$.

$$L^{asc}(\mathbf{w}) = L\left(\mathbf{w} + \rho \frac{\nabla L(\mathbf{w})}{||\nabla L(\mathbf{w})||_2}\right) - L(\mathbf{w}) \tag{3}$$

$$L^{avg}(\mathbf{w}) = \mathbb{E}_{\epsilon \sim \mathcal{N}(0, \max_j |w_j| \sigma_p^2 \mathbb{I})}[L(\mathbf{w} + \epsilon)] - L(\mathbf{w}) \tag{4}$$

# 4 UNDERSTANDING THE MECHANISMS OF TRAINING-INDUCED NOISE RESILIENCE

In this section, we study both SAM and RWP to understand how these methods arrive at minima robust to noise. We evaluate both methods across a variety of test-time noise settings while studying the dynamics they exhibit during training.

## 4.1 EXPERIMENTAL SETTING

We perform our experiments using the Cifar-100, Tiny-ImageNet and ImageNet-100 datasets Krizhevsky & Hinton (2009); Deng et al. (2009); Le & Yang (2015) with a vareity of model architectures. The results in the following sections are taken using a ResNet-18 backbone He et al. (2015) trained on Cifar-100; additional experiments with more backbone models/datasets are included in Appendix Sec. D and Sec. E. For each experimental result, we report two uncertainty measures: noise uncertainty (first number in tables), the standard deviation calculated across different samples of test-time noise, and weight uncertainty (second number in tables), the standard deviation calculated across different sets of model weights trained using a varying random seed. We average our results across 10 different weight-noise samples and 3 different trained model weights.

## 4.2 OBSERVING THE EVOLUTION OF NOISE-ROBUSTNESS DURING TRAINING

Before evaluating noise robustness on trained models, we first note that the perturbed distribution loss $L_{\mathcal{D}}^*$ does *not* follow the same trends of convergence as the unperturbed loss. To illustrate this,

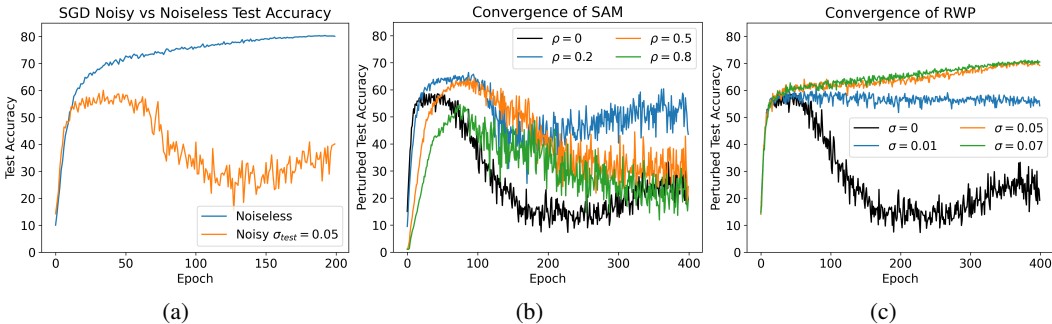

Figure 1: (a) ResNet-18 test accuracy on Cifar-100 as a function of training epoch when both perturbations ($\sigma_{test} = 0.05$) and no perturbations are applied. Training is conducted using SGD. (b) Comparison of perturbed test accuracy evolution for SAM with various $\rho$ values. (c) Comparison of perturbed test accuracy evolution for RWP with various $\sigma_{train}$ values.

we plot the test accuracy, both perturbed and unperturbed, of a ResNet-18 trained with SGD in Fig. 1a. Whereas the noiseless test accuracy continually increases until convergence, the perturbed test accuracy peaks at an earlier epoch, before gradually declining as the optimization continues. Intuitively, this matches with the hypothesis that in early training iterations, the loss landscape is naturally flat, and as such the perturbed test accuracy improves over the early epochs. However, without proper regularization, training will naturally begin to overfit as a sharper minimum is approached.

Next, we compare this same convergence dynamic for both SAM and RWP, varying perturbation strengths $\rho$ and $\sigma$, respectively. We visualize this comparison in Fig. 1b and Fig. 1c. In the models trained with SAM, we notice a similar dynamic to those trained with SGD: when trained for a sufficiently long schedule, the models undergo the same phenomenon of reaching a peak perturbed test accuracy, after which overfitting ensues. However, as $\rho$ is increased, two trends emerge: the epoch at which the peak perturbed accuracy is achieved at is pushed further back (i.e. the model can be trained closer to convergence before the peak is reached), and the perturbed accuracy at the peak is increased relative to that of SGD. After $\rho$ exceeds a critical value, these trends reverse: overfitting once again initiates at an earlier training epoch, and the peak accuracy is now reduced.

Meanwhile, the training dynamics observed in models trained with RWP stand in contrast to those trained with SAM and SGD. We consider two regimes: small $\sigma_{train}$, and large $\sigma_{train}$. In the former case, the optimization follows the same trend as in SGD and SAM: perturbed test accuracy peaks before convergence, after which further training leads to a decline in noise-resilience. However, in the latter case, the perturbed test accuracy does not peak, instead gradually increasing until the model converges. Even when the model is trained for a greater number of iterations beyond convergence, the perturbed test accuracy simply plateaus, and the model never enters an overfit regime. This stands in particular contrast to SAM, which *cannot* prevent the convergence of the model to an overfit solution, regardless of the selected perturbation radius. We defer a more detailed discussion of this phenomenon to Appendix Sec. B.1.

### 4.3 Selecting Training Perturbation Strength: Over-regularization is Optimal

Because RWP's training objective matches the distribution loss we are seeking to minimize, it is a natural choice to use it as regularization during training. Given a $\sigma_{test}$ applied to perturbations in the distribution loss, it is intuitive to select training-time variance $\sigma_{train}$ such that $\sigma_{train} = \sigma_{test}$. To test this hypothesis, we evaluate models trained using differing values of $\sigma_{train}$ on the validation dataset for selected values of $\sigma_{test}$, with our results shown in Tab. 1. In doing so, we observe a surprising effect: *over*-regularizing RWP training by selecting $\sigma_{train} > \sigma_{test}$ achieves better test accuracy on the held-out data. This suggests that the perturbations sampled from the tail of the noise distribution have a disproportionate effect on guiding the optimization to an optimally-flat minimum.

Next, we evaluate SAM training with a variety of $\rho$ values as shown in Tab. 2. As is the case with RWP, the SAM-trained models consistently outperform those trained with SGD on the perturbed

Table 1: Comparison of RWP-trained ResNet-18 on Cifar-100 with varying perturbation strength at both training and test time. Staircase indicates boundary between $\sigma_{train} \leq \sigma_{test}$ and $\sigma_{train} > \sigma_{test}$.

| | $\sigma_{test} = 0.0$ | $\sigma_{test} = 0.02$ | $\sigma_{test} = 0.03$ | $\sigma_{test} = 0.05$ | $\sigma_{test} = 0.07$ |
|---|---|---|---|---|---|
| $\sigma_{train} = 0.0$ | **79.31** $\pm 0.04$ | $77.24 \pm 0.51 \pm 0.23$ | $71.95 \pm 3.05 \pm 1.06$ | $56.21 \pm 2.34 \pm 1.38$ | $47.09 \pm 2.54 \pm 3.64$ |
| $\sigma_{train} = 0.02$ | $78.86 \pm 0.41$ | $77.64 \pm 0.25 \pm 0.36$ | $75.77 \pm 0.43 \pm 0.33$ | $67.11 \pm 1.61 \pm 0.69$ | $50.12 \pm 2.59 \pm 2.89$ |
| $\sigma_{train} = 0.03$ | $78.96 \pm 0.28$ | $77.57 \pm 0.23 \pm 0.26$ | $75.82 \pm 0.47 \pm 0.30$ | $67.97 \pm 1.38 \pm 0.58$ | $52.57 \pm 2.43 \pm 0.99$ |
| $\sigma_{train} = 0.05$ | $78.69 \pm 0.12$ | **77.65** $\pm 0.26 \pm 0.05$ | **76.26** $\pm 0.47 \pm 0.11$ | $70.43 \pm 1.26 \pm 0.16$ | $56.55 \pm 3.29 \pm 0.35$ |
| $\sigma_{train} = 0.06$ | $78.04 \pm 0.39$ | $77.15 \pm 0.27 \pm 0.26$ | $75.97 \pm 0.42 \pm 0.20$ | **71.04** $\pm 0.91 \pm 0.15$ | $59.27 \pm 2.50 \pm 0.20$ |
| $\sigma_{train} = 0.07$ | $77.17 \pm 0.34$ | $76.32 \pm 0.21 \pm 0.20$ | $75.18 \pm 0.37 \pm 0.23$ | $70.51 \pm 1.04 \pm 0.25$ | $59.39 \pm 2.64 \pm 0.10$ |
| $\sigma_{train} = 0.08$ | $77.05 \pm 0.32$ | $76.24 \pm 0.26 \pm 0.17$ | $75.15 \pm 0.45 \pm 0.18$ | $70.96 \pm 1.20 \pm 0.45$ | **61.36** $\pm 2.85 \pm 0.90$ |

Table 2: Comparison of SAM-trained ResNet-18 on Cifar-100 with varying perturbation strength at both training and test time.

| | $\sigma_{test} = 0.0$ | $\sigma_{test} = 0.02$ | $\sigma_{test} = 0.03$ | $\sigma_{test} = 0.05$ | $\sigma_{test} = 0.07$ |
|---|---|---|---|---|---|
| $\rho = 0.0$ | $79.31 \pm 0.04$ | $77.24 \pm 0.51 \pm 0.23$ | $71.95 \pm 3.05 \pm 1.06$ | $56.21 \pm 2.34 \pm 1.38$ | $47.09 \pm 2.54 \pm 3.64$ |
| $\rho = 0.2$ | **81.15** $\pm 0.21$ | $78.71 \pm 0.33 \pm 0.15$ | $75.27 \pm 1.89 \pm 1.71$ | $64.36 \pm 1.94 \pm 0.70$ | **53.61** $\pm 2.61 \pm 0.68$ |
| $\rho = 0.3$ | $80.27 \pm 0.19$ | **79.12** $\pm 0.23 \pm 0.28$ | **76.82** $\pm 1.13 \pm 1.09$ | **66.51** $\pm 2.57 \pm 1.27$ | $52.93 \pm 3.51 \pm 1.46$ |
| $\rho = 0.5$ | $78.34 \pm 0.05$ | $77.31 \pm 0.25 \pm 0.23$ | $74.41 \pm 2.00 \pm 0.97$ | $63.27 \pm 2.57 \pm 1.81$ | $47.16 \pm 4.19 \pm 2.35$ |
| $\rho = 0.8$ | $72.34 \pm 0.37$ | $71.04 \pm 0.30 \pm 0.46$ | $66.98 \pm 1.67 \pm 2.44$ | $50.85 \pm 3.26 \pm 3.82$ | $19.02 \pm 3.39 \pm 2.54$ |

accuracy metric. Although less straightforward to judge, we posit that as in the case of RWP, strongly-regularizing SAM through increasing $\rho$ should produce optimally-robust models. Comparing these experiments to the application of SAM in the standard setting of improving generalization without noise, we indeed find that the optimal perturbation radius is increased ($\rho = 0.2$ vs. $\rho = 0.3$). However, a surprising trend emerges when ramping test-time noise: the optimal value of $\rho$ is consistent across all of the test-time noise settings. Notably, models trained with $\rho = 0.8$, our most strongly perturbed SAM models, exhibit no performance advantage over those trained with SGD across any of the noise settings. For larger $\sigma_{test}$, a small value of $\rho$ cannot achieve the over-regularizing effect seen to be beneficial in RWP. The fact that SAM with larger perturbation radii fails to produce more noise-resilient models indicates that although stronger perturbations are needed for regularization, large worst-case perturbations have a deleterious effect on the convergence of training.

To capture the trade-off between a deeper vs flatter minimum, we plot $\sigma_{test} = 0.07$ test accuracy vs. unperturbed test accuracy for a variety of SGD, SAM and RWP models trained using varying $\sigma_{train}/\rho$ (Fig. 2a). In the case of RWP, we see that both deep-but-sharp and flat-but-shallow (the right and left regions respectively) perform non-ideally, with a modest region in the middle representing the optimal trade-off. For SAM, we see that in contrast to RWP, the most noise-resilient model is simultaneously the model with the largest unperturbed test accuracy, indicating that SAM *cannot* effectively increase flatness at the cost of the loss value. Additionally, we plot a comparison of the optimal SAM and RWP performance across a variety of test noise settings, shown in Fig. 2b. When the model weights are unperturbed at test time, SAM outperforms RWP, in agreement with the current understanding of SAM. For small-magnitude noise, SAM likewise maintains an advantage over RWP. However, as the perturbation strength is increased, the performance of RWP quickly overtakes SAM, with the performance gap further widening for greater noise. This result logically follows from our previous observation: although small-$\rho$ SAM is successful at inducing noise-robustness (and is in fact preferable depending on the test noise strength), training with large $\rho$ degrades the optimization, preventing SAM from scaling to more heavily-perturbed settings. Therefore, even in large $\sigma_{test}$ noise settings, small $\rho$ is optimal, but performs worse than optimal-$\sigma_{train}$ RWP.

## 4.4 DRAWBACK OF LARGE PERTURBATIONS: SHARPNESS-INDUCED VANISHING GRADIENT

From the results earlier in this section, we can deduce that strong adversarial perturbations are disruptive to the training process. To understand this effect, we plot the $l_2$-norm of the update gradient $||\nabla L||_2$ (i.e. the gradient at the perturbed point) as a function of training epoch (averaged over minibatches) for both techniques, shown in Fig. 3a. For fair comparison, we select SAM $\rho = 0.3$ and RWP $\sigma_{train} = 0.08$ as the representative trajectories to plot, as SAM $\rho = 0.3$ is optimal for most of the test-time noise settings and $\sigma_{train} = 0.08$ is the most-perturbed RWP model we train. In early epochs, we see the norm of the gradient increase as the training path enters into a steep valley within the loss landscape. Later, as training begins to converge to a local minimum, the gradient

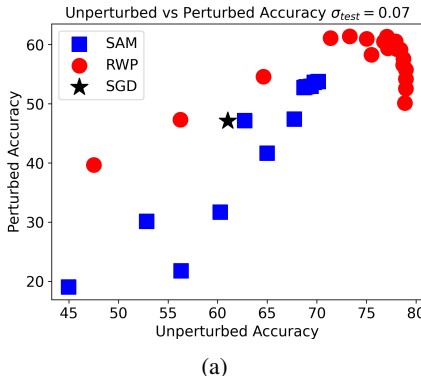
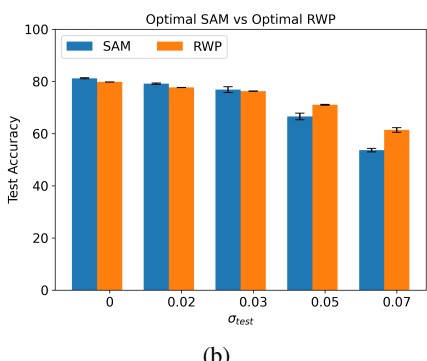

(a)                                        (b)

Figure 2: (a) Scatter plot of unperturbed test accuracy vs $\sigma_{test} = 0.07$ perturbed test accuracy for varying perturbation sizes of SAM, RWP and SGD. (b) Comparison between optimal RWP and SAM across a variety of noise settings.

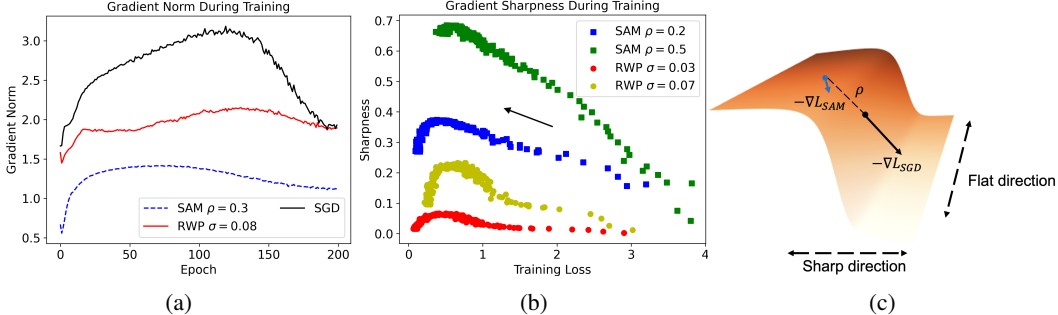

(a)                                (b)                                (c)

Figure 3: (a) Plot of the update gradient norm $||\nabla L||_2$ as a function of training epoch for a ResNet-18 trained on Cifar-100 using SGD, SAM, and RWP. (b) Plot of the gradient sharpness (corresponding to ascent-direction for SAM or average-direction for RWP) of both SAM and RWP as a function of training loss. (c) Schematic visualization of a loss surface, demonstrating a *sharp* direction (along which the SAM perturbation lies), and a *flat* direction (along which the RWP perturbations lie). In sharpness plot, arrow indicate direction of increasing training epochs.

norm decreases again. Relative to SGD, all of the perturbative training methods reduce the gradient norm across the entire optimization path. However, we notice a stark contrast: the SAM-trained model produce gradient norms that are *significantly* more reduced in magnitude than those of the RWP training. Taking note that RWP's perturbation magnitude is two orders of magnitude larger than that of SAM (63 vs 0.3), this implies that the second directional derivative is *markedly* larger in magnitude along the direction of the gradient than in a randomly sampled direction. This issue of the vanishing gradient is further exacerbated for adversarial perturbations of greater $\rho$ value; beyond a certain point, the gradient is too small in magnitude for learning to occur, and training fails to progress meaningfully beyond the initialized state.

Given this observation, we next build an intuitive interpretation to understand why SAM perturbations reduce the gradient norm to a greater degree than RWP perturbations. To aid in this understanding, we visualize the gradient sharpness during training to understand the loss topography at the perturbed point, shown in Fig. 3b. From this, we see that the sharpness encountered by the small SAM perturbations is much larger in magnitude than that encountered by the large RWP perturbations, a trend that holds true at every point within the loss landscape. This strong sharpness combined with the previously discussed diminishing gradient norm suggests that, along the gradient direction, the loss landscape contains a steep ridge in which both the loss and the loss gradient are rapidly changing and beyond which lies a largely-flat high-loss plateau. While small SAM perturbations produce gradients along the steeper incline of the ridge, scaling $\rho$ beyond this point perturbs the weights onto the flat plateau, where gradients are vanishingly-small. On the other hand, random

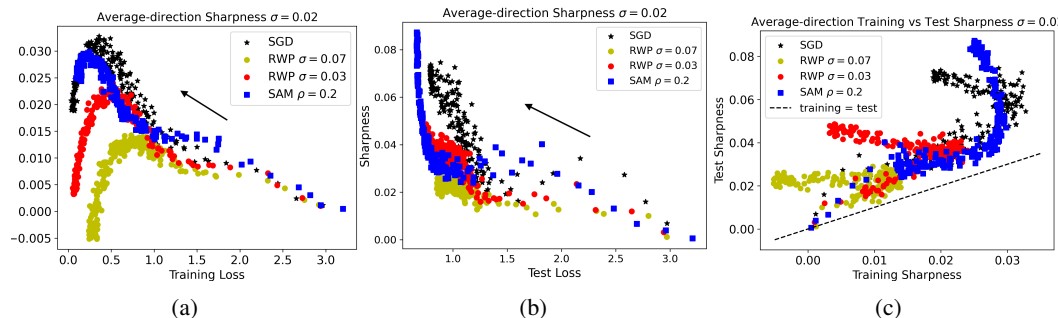

Figure 4: (a) Plot of ResNet-18 Cifar-100 average-direction sharpness for $\sigma = 0.02$. as a function of training loss. (b) Plot of average-direction sharpness as a function of *test* loss for $\sigma = 0.02$. (c) Plot of *test* sharpness as a function of *training* for $\sigma = 0.02$.

perturbations, even of significant perturbation radii, encounter a drastically smaller increase in the loss value and a drastically smaller decrease in the gradient magnitude, in-line with empirical observations of flatness in most directions of the loss landscape Li et al. (2018). As a result, perturbations along these directions produce points still within the steep low-loss basin, and as such do not encounter vanishing gradients to the same degree as those along the gradient direction. To visually capture this intuitive understanding of the landscape, we include a diagram in Fig. 3c.

Combining our previous observation that over-regularization benefits training with the trend toward a vanishing gradient, we conclude that there exists a *critical perturbation length*, which we define informally as the perturbation length, present in both SAM and RWP, at which point the harm from the diminishing gradient overtakes the benefit of over-regularization, producing a minimum *less* optimally noise-robust.

### 4.5 DE-CORRELATING TRAINING AND TEST SHARPNESS

In adopting SAM and RWP as valid methods for improving noise-robustness, we implicitly make the assumption that a flat *training* loss landscape necessarily correlates with a flat *test* loss landscape. To empirically test this hypothesis, we plot average-direction training sharpness as a function of training loss, average direction test sharpness as a function of test loss, and test sharpness as a function of training sharpness ($\sigma = 0.02$) for several SAM and RWP training trajectories, displayed in Fig. 4. Doing so, we observe several unintuitive trends regarding the correlation between training and test sharpness. First, we note that a mininum's test sharpness is almost always greater than its training sharpness, made especially clear in Fig. 4c. Second, we note that, particularly toward the later training epochs, the evolution of training and test sharpness are in fact anti-correlated (i.e. test sharpness *increases* as training sharpness *decreases*). Lastly, we observe a surprising difference in the generalization of SAM and RWP's minima, respectively: although RWP $\sigma = 0.03$ locates a minimum both *deeper* and *flatter* in the training landscape than that of SAM, SAM's minimum generalizes to a lower test loss and as such is more noise-resilient. This result firmly establishes a new paradigm for training noise-robust models: locating a point displaying flatness in the test loss landscape should be considered independently of the same problem in the training loss landscape.

## 5 MITIGATING THE VANISHING GRADIENT: DYNAMIC PERTURBATION SCHEDULES

In Sec. 4, we hypothesize the existence of a critical perturbation length, denoting the maximal strength of perturbation before the vanishing gradient effect begins to dominate, reducing noise robustness. In this section, we examine the question of *how* this critical perturbation evolves as training progresses, and we develop dynamic SAM and RWP perturbation schedules to match this evolution.

Table 3: Comparison of our three proposed perturbation schedules for SAM/RWP on ResNet-18 Cifar-100 training. The maximum perturbation strength/schedule length are determined through using a grid search.

|  | $\sigma_{test} = 0.02$ | $\sigma_{test} = 0.03$ | $\sigma_{test} = 0.05$ | $\sigma_{test} = 0.07$ |
|---|---|---|---|---|
| Constant-$\sigma$ RWP | $77.65 \pm 0.26 \pm 0.05$ | $76.26 \pm 0.47 \pm 0.11$ | $71.04 \pm 0.91 \pm 0.15$ | $61.36 \pm 2.85 \pm 0.90$ |
| Linear-$\sigma$ RWP | $77.95 \pm 0.26 \pm 0.17$ | $76.61 \pm 0.44 \pm 0.13$ | $71.85 \pm 0.90 \pm 1.23$ | $62.24 \pm 3.52 \pm 1.39$ |
| Quadratic-$\sigma$ RWP | $\mathbf{78.26} \pm 0.20 \pm 0.13$ | $\mathbf{77.00} \pm 0.30 \pm 0.17$ | $\mathbf{72.43} \pm 0.82 \pm 0.21$ | $\mathbf{65.27} \pm 2.05 \pm 0.64$ |
| Off-to-On RWP | $78.00 \pm 0.30 \pm 0.09$ | $76.63 \pm 0.44 \pm 0.14$ | $72.31 \pm 1.03 \pm 0.50$ | $63.15 \pm 2.84 \pm 1.61$ |
| Constant-$\rho$ SAM | $79.12 \pm 0.23 \pm 0.28$ | $76.82 \pm 1.13 \pm 1.09$ | $66.51 \pm 2.57 \pm 1.27$ | $53.61 \pm 2.61 \pm 0.68$ |
| Linear-$\rho$ SAM | $79.40 \pm 0.21 \pm 0.19$ | $\mathbf{77.61} \pm 0.51 \pm 0.42$ | $67.27 \pm 1.18 \pm 1.10$ | $58.37 \pm 2.42 \pm 0.75$ |
| Quadratic-$\rho$ SAM | $\mathbf{79.49} \pm 0.34 \pm 0.14$ | $77.13 \pm 1.52 \pm 0.56$ | $\mathbf{68.11} \pm 1.10 \pm 0.29$ | $\mathbf{60.24} \pm 1.98 \pm 0.83$ |
| Off-to-On SAM | $79.16 \pm 0.27 \pm 0.14$ | $76.86 \pm 0.82 \pm 0.44$ | $65.12 \pm 1.28 \pm 0.87$ | $56.10 \pm 3.08 \pm 1.96$ |

## 5.1 BROADENING OF THE LOSS LANDSCAPE

From our previous experiments, we know that the critical perturbation length is closely tied to the proximity of the perturbed point to the high-loss, flat plateau of the loss landscape. However, as training proceeds, the shape of the loss basin in the immediate neighborhood likewise changes, suggesting that this proximity will *not* remain constant for a given perturbation size. To visualize this effect, we track the evolution of the perturbed training loss (i.e. the training loss evaluated at the perturbed set of weights $L(w_p)$) in Appendix Fig. 12, allowing us to better localize where the update gradient is calculated at within the loss landscape relative to the high-loss plateau. For all but the largest-magnitude perturbations, we observe that the perturbed loss gradually decreases as a function of the training loss until convergence. This indicates a broadening of the loss basin: as training proceeds, a larger perturbation is required to enter the high-loss plateau, and as such the loss at a constant perturbation distance consistently decreases.

Based on this observation, we contend that the critical perturbation length should also increase alongside the width of the loss basin, suggesting that the optimal perturbation strength selected at the initial stage of training should not remain constant. This insight leads us to develop modified training schedules for SAM and RWP, which we introduce in the following subsection.

## 5.2 MATCHING THE OBSERVED LANDSCAPE EVOLUTION: EMPIRICALLY-SELECTED PERTURBATION SCHEDULES

With an established understanding that the critical perturbation length increases during training, this opens the path to more-strategic over-regularization: instead of statically selecting a training perturbation magnitude that need be stable for the entire duration of training, instead we aim to engineer the training perturbations to match the evolution of the dynamic loss landscape. From our observations, we know that this perturbation magnitude should continually increase in magnitude throughout training, however the exact manner in which the critical perturbation length evolves is not clear. As such, we propose exploring a variety of differing perturbation schedules, empirically selecting the optimal strategy based on noise-robustness. Specifically, we propose three varieties of modified training schedules: a **linear** ramp, **quadratic** ramp, and a hard **off-to-on**. All three schedules initialize perturbation strength at 0, and increase to a terminal value over the course of a warm-up cycle.

We perform experiments using all three schedule variations with a ResNet-18 model and compare across a variety of noise settings, shown in Tab. 3. From these results, we see that across all of the test-time noise settings, the ramped perturbation schedules outperform the constant-perturbations baselines of both SAM and RWP. Of the three schedule varieties, the quadratic ramp schedule performs best across most of the scenarios, suggesting the true evolution of the critical perturbation length follows a similar trajectory. We also observe that the accuracy gain achieved by applying the dynamic perturbation schedule *increases* as $\sigma_{test}$ (and hence $\sigma_{train}$ or $\rho$) is increased. This aligns with our understanding of the critical perturbation length: for larger $\sigma_{test}$, larger perturbations are required to achieve an over-regularizing effect, but are also further inhibited by proximity to the critical perturbation length. By applying an increasing ramp schedule, far larger perturbations than otherwise possible can be tolerated, producing the most benefit in the large $\sigma_{test}$ scenario.

Table 4: Comparison of SGD, SAM and RWP-trained ResNet-18 when performing simulated inference on analog accelerators.

|  | RRAM | SONOS |
|---|---|---|
| SGD | $65.61 \pm 2.68 \pm 0.51$ | $79.31 \pm 0.03 \pm 0.02$ |
| SAM $\rho = 0.4$ | $68.42 \pm 1.25 \pm 0.89$ | $80.07 \pm 0.03 \pm 0.22$ |
| SAM + Quadratic Schedule | $66.16 \pm 1.52 \pm 2.58$ | $\mathbf{80.84} \pm 0.03 \pm 0.09$ |
| RWP $\sigma = 0.05$ | $73.96 \pm 0.90 \pm 0.38$ | $78.69 \pm 0.02 \pm 0.12$ |
| RWP + Quadratic Schedule | $\mathbf{74.33} \pm 0.88 \pm 0.12$ | $78.89 \pm 0.03 \pm 0.17$ |
| Device-Aware Training (RRAM) | $69.35 \pm 1.29 \pm 0.35$ | - |
| Device-Aware Training (SONOS) | - | $72.51 \pm 0.03 \pm 0.29$ |

To confirm that increased perturbation strength is now feasible, we perform ablations on two hyperparameters of our adjustable schedules: the maximal perturbation strength after ramping and the ramping duration. Using a grid search (Appendix Tab. 11, 12), we find that the optimal max perturbations are higher than those in constant perturbation experiments (Tab. 1, 2). Notably, SAM's optimal $\rho$ increases significantly from 0.3 to 1.0, aligning better with the dynamic loss landscape and supporting our hypothesis of an increasing critical perturbation length. This demonstrates that greater over-regularization is achievable when introduced at the appropriate training stage, and opens a path toward perturbative training which is engineered around the evolution of the loss landscape.

## 6 Validation through Analog Hardware Simulation

In this section, we demonstrate our results for a practical use-case by simulating model inference on analog hardware accelerators. To perform these experiments, we leverage CrossSim Feinberg et al., an open-source python-based simulator that models AIMC in the context of applications reliant on matrix multiplication (e.g. neural networks). We select two well-known memory devices (both supported in CrossSim) for our inference modeling: RRAM (resistive random-access memory) Milo et al. (2021) and SONOS (silicon-oxide- nitride-oxide-silicon) Xiao et al. (2022a). Both devices exhibit programming error profiles that are approximately state-independent, as is the case in our perturbation experiments. Whereas the error of SONOS empirically follows a small-$\sigma$ distribution, RRAM produces significantly larger magnitude errors; to quantify this, we calculate the RMSE between the totality of the perturbed/unperturbed model weights, which we find to be $23.936 \pm 0.004$ for RRAM, and $2.371 \pm 0.0005$ for SONOS. In addition to evaluating several of the ResNet-18 models trained in the previous sections, we also train models using CrossSim to directly inject perturbations matching the hardware's noise profile during training, allowing us to test the hypothesis of unmatched training/test distributions. We compare these results in Tab. 4. From the table, we see that the general trends observed previously hold true: both SAM and RWP improve noise-robustness relative to standard SGD on the simulated hardware. On the small-noise SONOS hardware, SAM outperforms RWP, while on the large-noise RRAM, RWP outperforms SAM. In both cases, a further performance improvement is achieved through the adoption of a quadratic perturbation schedule. Additionally, optimal RWP/SAM produce better-performing models than directly applying the hardware noise during training, emphasizing again that simply matching training and test-time regularization *does not* produce optimal noise-robustness.

## 7 Conclusion

In this paper, we conduct a comprehensive study on the use of flatness-finding optimization methods, namely SAM and RWP, for finding neural network minima robust to perturbations in weight-space. First, we identify general trends in the application of these techniques to finding noise-robust minima, and discover that over-regularized training in the form of strong perturbations produces the most robust weights. At the same time, we identify a deleterious effect of strong perturbations: a vanishing-gradient effect induced by sharpness in the loss landscape, particularly pronounced in SAM. To curtail this effect, we introduce a ramped perturbation schedule, in which perturbation magnitude is gradually increased as training progresses, allowing for perturbations to evolve naturally with the widening loss basin. We hope that this extensive investigation of training neural networks robust to weight noise spurs future research of this under-explored problem.

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

## A    Implementation Details

All Cifar-100/Tiny-ImageNet models are trained for 200 epochs using SGD as the base optimizer. We use an initial learning rate of $0.05$ with a cosine learning rate schedule. We use a weight decay of $5 * 10^{-4}$ and momentum of $0.9$. We apply the standard data augmentations of random cropping, flipping, and normalization. In addition, we train using label smoothing of 0.1. For the ImageNet-100 models, we train for 100 epochs using an initial learning rate of 1 and weight decay of $1 * 10^{-4}$. In all experiments, early stopping is applied to select the training epoch which achieves the highest perturbed test accuracy.

## B    Detailing the Dynamics of Perturbed Training

### B.1    Understanding SAM's Overfitting

To empirically investigate the evolution of SAM's noise robustness during training, we plot the cosine similarity between the original loss gradient $\nabla L(\mathbf{w})$ and the perturbed gradient $\nabla L(\mathbf{w} + \epsilon)$ for both SAM and RWP, shown in Fig. 5. From this, we see that for RWP, the cosine similarity between the gradients gradually diverges as training progresses and the optimization path enters a deeper loss valley with more rapidly-changing gradients. As such, the gradients are, on average, nearly orthogonal by the end of training. On the other hand, in the SAM-trained models, the cosine similarity drops sharply after the early training iterations, but thereafter roughly plateaus for the rest of the training schedule. As a result, even as the training converges to a sharper minimum, the SAM gradient still contains a *significant* component along the direction of the original gradient. Hence, further training will continue to push the model toward a minimum more characteristic of the one found by SGD, although at an attenuated rate. For the strongly perturbed SAM using $\rho = 1$, the gradients do diverge at later training epochs; however the strong perturbations in early iterations prevent the optimization from progressing meaningfully, as will be shown in the following section.

### B.2    Charting SAM/RWP Performance in Low-Noise Regime

In addition to Fig. 2a, we also plot the $\sigma_{test} = 0.02$ perturbed accuracy as a function of unperturbed accuracy, shown in Fig. 6. In this case, we observe that SAM and RWP follow a similar linear trend, with the exception that the lower-loss SAM minima achieve greater noise-resiliency than the best RWP minima.

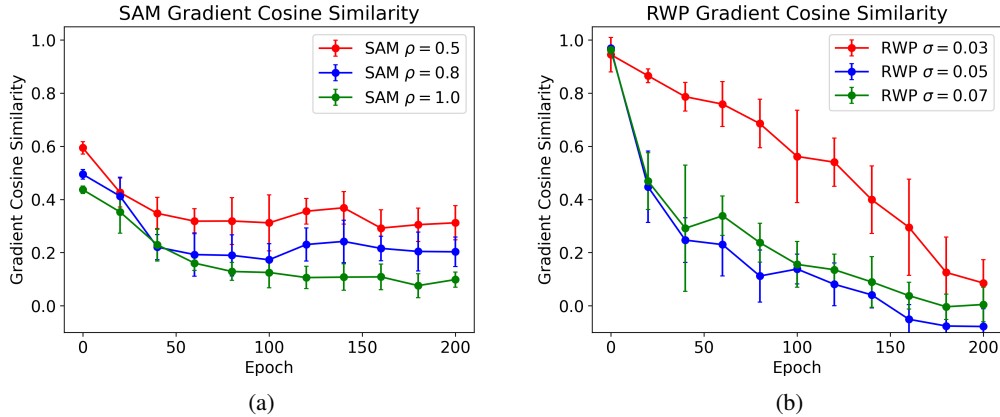

Figure 5: Plot of cosine similarity between perturbed/unperturbed gradients for various (a) SAM and (b) RWP configurations on a ResNet-18 as a function of training epoch.

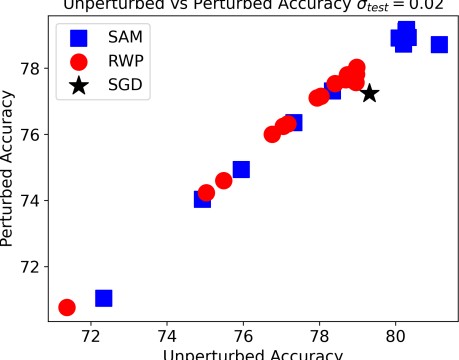

Figure 6: Scatter plot of unperturbed test accuracy vs $\sigma_{test} = 0.02$ perturbed test accuracy for a variety of ResNet-18 models trained using SAM, RWP and SGD on Cifar-100.

### B.3 VISUALIZING THE LOSS LANDSCAPE

To visually analyze the training loss minima found through perturbative training, we produce plots visualizing the loss landscape (both training and test) of several different minima (using the method proposed in Li et al. (2018)), shown in Fig. 7. In the top row of the figure, (e.g. Figs. 7a, 7b, 7c), we visualize models trained with RWP with increasing $\sigma_{train}$. As expected, the model trained using the larger value of $\sigma_{train}$ produces a visually much flatter minimum, confirming the straightforward correlation we expect. This same pattern holds true for the test landscape minima of RWP, plotted in Figs. 7g, 7h, 7i. Comparing SAM's training minima amongst each other, we see that $\rho = 0.3$ appears the flattest, in agreement with our quantitative results. However, comparing the SAM minima against RWP minima breaks the straightforward correlation between training/test loss flatness. For example, the RWP $\sigma_{train} = 0.05$ training minimum (Fig. 7b) is flatter than all of the SAM minima (Fig. 7d,7e,7f); however the test landscape minima for SAM $\rho = 0.2$ and $\rho = 0.3$ (Fig. 7j, 7k) are both flatter than the corresponding test minimum for RWP (Fig. 7h). This agrees with our previous experimental results: although sharper from the perspective of the training landscape, SAM models outperform RWP in the small $\sigma_{test}$ noise settings. These visualizations provide further evidence that the relative training loss flatness and test loss flatness need not correlate.

### B.4 QUANTIFYING ATTENUATED TRAINING

To confirm the effect of the sharpness-induced vanishing gradient during training, we plot the gradient norm during training alongside the total distance traveled in the parameter space as training unfolds $\sum_{t=1}^{N} ||w_{t+1} - w_t||$ (where $t$ corresponds to epoch number), shown in Fig. 8b. Compared to SGD, we see that all of the perturbed training trajectories display both a smaller gradient norm and smaller distance traveled. For the case of SAM, we see a dramatically smaller distance traveled as compared to the RWP models, demonstrating that the diminished gradient magnitude indeed attenuates training.

### B.5 COMPARING TRAINING AND TEST SHARPNESS DURING TRAINING

In addition to the plots in Fig. 4, we also compare the average-direction training/test sharpness for $\sigma = 0.07$, plotted in Fig. 9. As was previously shown, we find a notable divergence between a minimum's training loss sharpness and test loss sharpness. For example, we see in the plot of training sharpness (Fig. 9a) that SAM exhibits virtually the same degree of flatness as SGD over its entire trajectory. However, when plotting the test sharpness of the same trajectories, we see that in the earlier epochs SAM traverses a test loss basin that is significantly less sharp than that of SGD; it is within this basin that SAM (with early stopping) finds weights with improved noise-resilience.

## C DOES NOISE DISTRIBUTION MATTER?

In all of our previous analysis, we have made the assumption of matching RWP's training noise distribution class with that of our test noise distribution $\mathbb{Q}$, in this case a normal distribution. We note that this assumption grants RWP with knowledge at training time of the shape of the test distribution, an advantage that SAM does not benefit from. To decouple the flatness-finding capability of RWP from its a priori knowledge of the test distribution, we perform experiments in which we define $\mathbb{Q} \sim Laplace(0, b \max_i |w_i|)$, allowing us to evaluate noise-robustness in an unbiased manner. We present these results in Tab. 5. Even in the case of Laplace noise, we observe a similar trend as in Gaussian noise at test time: SAM outperforms RWP for lower-strength noise, but cannot compensate when the noise magnitude is increased. From these results, we conclude that RWP is indeed locating broadly-flat minima which are generally robust against random corruptions.

## D EXPERIMENTS ON DIFFERING BACKBONE MODELS

### D.1 EVALUATING GENERALTIY OF SAM/RWP PERFORMANCE

To confirm that our observations of SAM/RWP hold true generally, we perform similar experiments as in Sec. 4.3 on a variety of ResNets and WRNs, shown in Tab. 6 and Tab. 7. For a fair comparison, we replace the bottleneck block in the ResNet-50 with the same basic block architecture used in

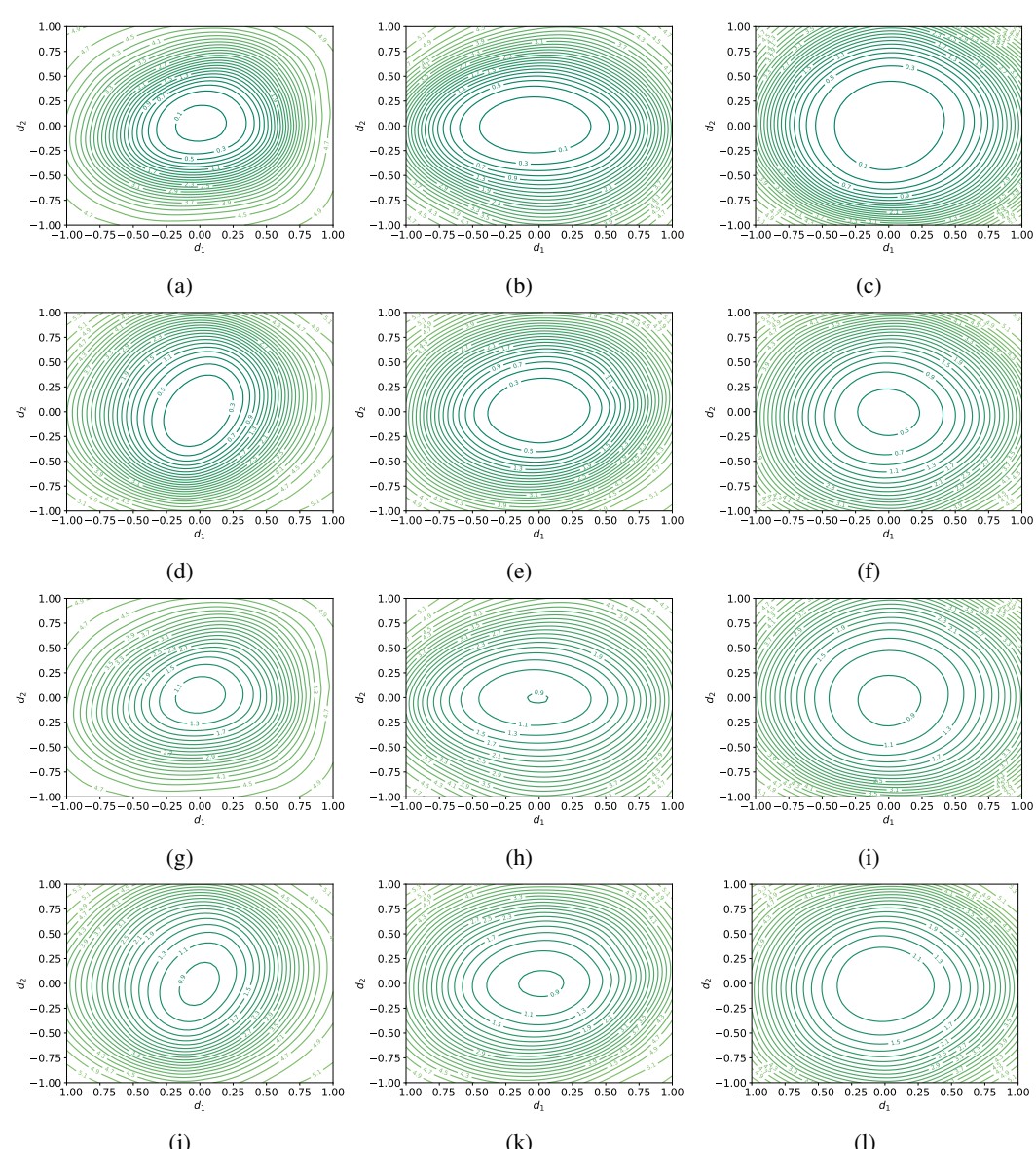

Figure 7: 2D visualization along filter-normalized directions of the (unperturbed) Cifar-100 loss landscape of minima found using RWP $\sigma_{train} = 0.02$, $\sigma_{train} = 0.05$, $\sigma_{train} = 0.07$ ((a)-(c) for training landscape visualization, (g)-(i) for test landscape visualization), and SAM $\rho = 0.2$, $\rho = 0.3$, $\rho = 0.5$ ((d)-(f) for training landscape visualization, (j)-(l) for test landscape visualization).

Table 5: Comparison on Cifar-100 of SGD, SAM and RWP-trained ResNet-18 when test-time noise follows a Laplace distribution $Laplace(0, b \max_i |w_i|)$.

|  | $b = 0.01$ | $b = 0.02$ | $b = 0.03$ | $b = 0.04$ |
|---|---|---|---|---|
| SGD | $78.49 \pm 0.18 \pm 0.07$ | $73.42 \pm 2.59 \pm 0.39$ | $59.66 \pm 1.74 \pm 1.22$ | $54.08 \pm 2.05 \pm 1.54$ |
| SAM $\rho = 0.2$ | $79.52 \pm 0.15 \pm 0.14$ | $77.05 \pm 0.38 \pm 0.16$ | $67.85 \pm 2.76 \pm 0.55$ | $61.11 \pm 1.63 \pm 1.40$ |
| SAM $\rho = 0.3$ | $\mathbf{79.75} \pm 0.14 \pm 0.23$ | $\mathbf{77.66} \pm 0.78 \pm 0.59$ | $69.47 \pm 1.60 \pm 0.97$ | $62.25 \pm 2.40 \pm 0.82$ |
| SAM $\rho = 0.5$ | $77.95 \pm 0.16 \pm 0.08$ | $75.41 \pm 2.02 \pm 0.78$ | $66.72 \pm 1.58 \pm 0.43$ | $58.35 \pm 5.19 \pm 2.01$ |
| RWP $\sigma = 0.03$ | $78.30 \pm 0.15 \pm 0.25$ | $76.32 \pm 0.42 \pm 0.27$ | $72.05 \pm 1.07 \pm 0.41$ | $63.18 \pm 2.71 \pm 1.18$ |
| RWP $\sigma = 0.05$ | $78.28 \pm 0.14 \pm 0.08$ | $76.66 \pm 0.35 \pm 0.05$ | $\mathbf{73.53} \pm 0.72 \pm 0.14$ | $67.40 \pm 1.66 \pm 0.34$ |
| RWP $\sigma = 0.07$ | $76.82 \pm 0.14 \pm 0.17$ | $75.56 \pm 0.34 \pm 0.13$ | $72.95 \pm 0.66 \pm 0.23$ | $\mathbf{68.00} \pm 1.53 \pm 0.64$ |

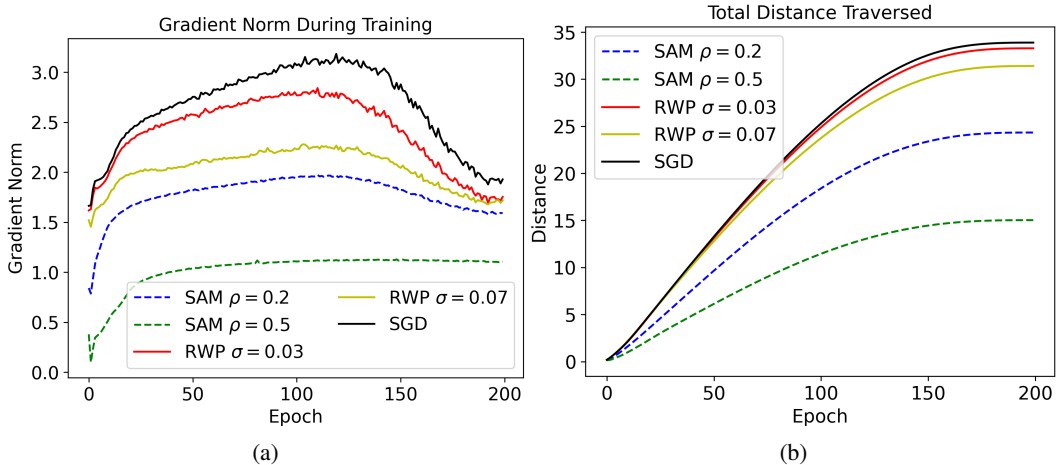

Figure 8: (a) Plot of the ResNet-18 update gradient norm $||\nabla L||_2$ as a function of training epoch for SGD, SAM, and RWP on Cifar-100. (b)Plot of the total distance traversed during training, $\sum_{i=1}^{N} ||w_{t+1} - w_t||$

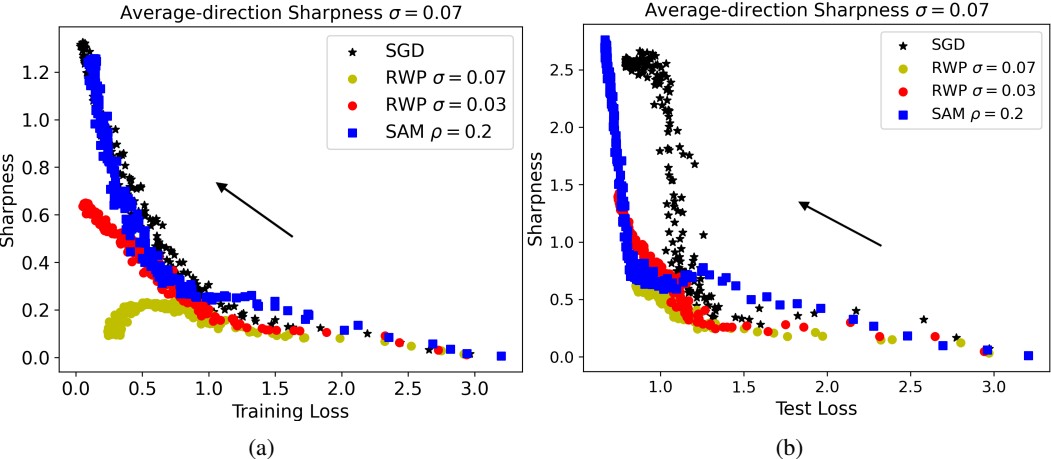

Figure 9: (a) Plot of average-direction sharpness for $\sigma_{test} = 0.02$ for a ResNet-18 training on Cifar-100. (b) Plot of average-direction sharpness as a function of *test* loss for $\sigma_{test} = 0.02$.

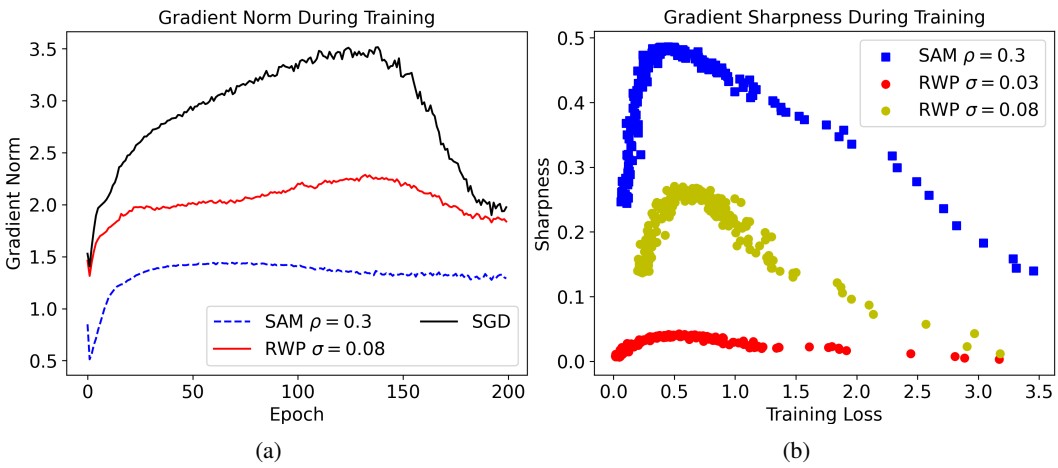

Figure 10: Plots of (a) gradient norm and (b) gradient sharpness for ResNet-50 during Cifar-100 training.

Table 6: Comparison of RWP and SAM for ResNets of varying depths. The best-performing optimizer for a given architecture is bolded, whereas the best-performing architecture for a given optimizer is underlined.

| | $\sigma_{test} = 0.02$ | | $\sigma_{test} = 0.07$ | |
|---|---|---|---|---|
| | SAM | RWP | SAM | RWP |
| ResNet-9 | $\mathbf{74.14} \pm 0.94 \pm 1.67$ | $72.70 \pm 0.31 \pm 0.07$ | $39.58 \pm 5.03 \pm 0.97$ | $\mathbf{51.10} \pm 3.14 \pm 1.70$ |
| ResNet-18 | $\mathbf{79.12} \pm 0.23 \pm 0.28$ | $77.65 \pm 0.26 \pm 0.05$ | $53.61 \pm 2.61 \pm 0.68$ | $\mathbf{61.36} \pm 2.85 \pm 0.90$ |
| ResNet-34 | $\mathbf{80.82} \pm 0.24 \pm 0.18$ | $79.19 \pm 0.21 \pm 0.19$ | $\underline{55.38} \pm 3.23 \pm 1.67$ | $\mathbf{64.67} \pm 2.54 \pm 0.54$ |
| ResNet-50 | $\mathbf{\underline{81.29}} \pm 0.17 \pm 0.17$ | $\underline{79.81} \pm 0.19 \pm 0.17$ | $54.22 \pm 3.39 \pm 2.85$ | $\mathbf{\underline{64.87}} \pm 2.28 \pm 1.23$ |

the shallower ResNets. As is the case with ResNet-18, we find that for $\sigma_{test} = 0.02$, SAM-trained models achieve higher accuracy, whereas for $\sigma_{test} = 0.07$, RWP-trained models dominate. We additionally recreate Fig. 3 for training with a ResNet-50, demonstrating the same vanishing gradient effect observed in the ResNet-18.

## D.2 DYNAMIC PERTURBATION SCHEDULES

To verify the generality of our dynamic perturbation schedules, we also perform experiments using different backbone models, specifically ResNet-50, WRN-16-10, and PyramidNet-50 Han et al. (2017) shown in Tab. 8. Here, we note a contrast between the WRN and the ResNet-50/PyramidNet-50: while the perturbation schedule enhances noise-robustness for both network architectures, the improvement is markedly larger for the latter two networks. We conclude that ResNet-50 and PyramidNet-50 naturally exhibit a sharper loss landscape that is a significant factor in the degraded performance of SAM/RWP; as a result, the ramped perturbations improve performance by a large amount. Meanwhile, the naturally smooth loss landscape of the WRN leads to a fairly stable training process, even without the perturbation schedules. As a result, the improvement from applying the

Table 7: Comparison of RWP and SAM for ResNets of varying widths. The best-performing optimizer for a given architecture is bolded, whereas the best-performing architecture for a given optimizer is underlined.

| | $\sigma_{test} = 0.02$ | | $\sigma_{test} = 0.07$ | |
|---|---|---|---|---|
| | SAM | RWP | SAM | RWP |
| ResNet-18 | $\mathbf{79.12} \pm 0.23 \pm 0.28$ | $77.65 \pm 0.26 \pm 0.05$ | $53.61 \pm 2.61 \pm 0.68$ | $\mathbf{61.36} \pm 2.85 \pm 0.90$ |
| WRN-16-5 | $\mathbf{80.75} \pm 0.31 \pm 0.11$ | $79.36 \pm 0.27 \pm 0.06$ | $51.92 \pm 4.79 \pm 0.69$ | $\mathbf{59.11} \pm 3.48 \pm 0.18$ |
| WRN-16-10 | $\mathbf{83.13} \pm 0.23 \pm 0.32$ | $81.08 \pm 0.21 \pm 0.28$ | $\underline{60.11} \pm 3.04 \pm 2.31$ | $\mathbf{\underline{66.23}} \pm 2.26 \pm 0.75$ |
| WRN-16-15 | $\mathbf{\underline{83.53}} \pm 0.18 \pm 0.09$ | $\underline{81.24} \pm 0.21 \pm 0.08$ | $59.55 \pm 3.78 \pm 2.38$ | $\mathbf{64.63} \pm 3.96 \pm 0.83$ |

Table 8: Comparison of perturbed test accuracy when perturbation schedules are applied during training for differing backbone architectures.

| | | $\sigma_{test} = 0.07$ | |
|---|---|---|---|
| | Schedule | SAM | RWP |
| ResNet-50 | Constant | $54.22 \pm 3.39 \pm 2.85$ | $64.87 \pm 2.28 \pm 1.23$ |
| ResNet-50 | Quadratic | $\mathbf{59.18} \pm 3.23 \pm 2.69$ | $\mathbf{69.35} \pm 1.29 \pm 0.35$ |
| WRN-16-10 | Constant | $60.11 \pm 3.04 \pm 2.31$ | $66.23 \pm 2.26 \pm 0.75$ |
| WRN-16-10 | Quadratic | $\mathbf{61.60} \pm 3.19 \pm 0.49$ | $\mathbf{67.19} \pm 2.28 \pm 0.90$ |
| PyramidNet-50 | Constant | $47.40 \pm 3.49 \pm 1.25$ | $58.89 \pm 3.23 \pm 2.71$ |
| PyramidNet-50 | Quadratic | $\mathbf{52.95} \pm 3.48 \pm 2.69$ | $\mathbf{63.88} \pm 1.73 \pm 1.22$ |

Table 9: Comparison of SGD, SAM and RWP-trained ResNet-18 on Tiny-Imagenet.

| | $\sigma_{test} = 0.02$ | $\sigma_{test} = 0.05$ |
|---|---|---|
| SGD | $62.89 \pm 0.65 \pm 0.37$ | $36.99 \pm 2.64 \pm 2.06$ |
| SAM | $64.50 \pm 1.03 \pm 0.56$ | $45.93 \pm 1.48 \pm 0.86$ |
| SAM + Quadratic Schedule | $\mathbf{66.01} \pm 1.84 \pm 1.56$ | $41.41 \pm 1.82 \pm 0.91$ |
| RWP | $64.17 \pm 0.44 \pm 0.46$ | $54.26 \pm 1.95 \pm 1.24$ |
| RWP + Quadratic Schedule | $65.73 \pm 0.40 \pm 0.18$ | $\mathbf{56.26} \pm 1.75 \pm 0.15$ |

schedule is much more modest, in-line with performance gains in the small-noise setting applied to the narrower networks.

# E    TINY-IMAGENET AND IMAGENET-100 EXPERIMENTS

As further evidence for the generality of our results, we perform additional experiments on both the intermediate-scale Tiny-ImageNet dataset and the large-scale ImageNet-100 dataset. First, we plot the gradient norm of and gradient sharpness evolution during training on both Tiny-ImageNet and ImageNet-100, shown in Fig. 11, observing similar evidence for the sharpness-induced vanishing gradient as was seen in the Cifar-100 experiments. Next, we compare the noise-robust performance of SGD, SAM and RWP across a variety of noise settings on Tiny-ImageNet, shown in Tab. 9 and Tab. 10, where we again observe similar trends in regards to the noise-robustness of SAM and RWP as in the case of Cifar-100.

# F    DYNAMIC SCHEDULE DETAILS

## F.1    MOTIVATING DYNAMIC SCHEDULES: PERTURBED LOSS EVOLUTION

To illustrate the evolution of the critical perturbation length, we plot the perturbed point loss for several SAM and RWP training trajectories, shown in Fig. 12. In the case of extreme perturbation magnitude beyond the critical perturbation length, the perturbed point consistently lies on the high-loss plateau, preventing the perturbed loss from decreasing as is shown in the yellow lines. Otherwise, we see that for a consistent perturbation strength, the perturbed point loss decreases as training proceeds, indicating that for a consistent perturbation radii, the perturbed point's distance from the

Table 10: Comparison of ResNet-18 trained with various perturbation strengths of SAM and RWP for several test noise settings on Tiny-ImageNet.

| | $\sigma_{test} = 0.0$ | $\sigma_{test} = 0.02$ | $\sigma_{test} = 0.03$ | $\sigma_{test} = 0.04$ | $\sigma_{test} = 0.05$ |
|---|---|---|---|---|---|
| $\sigma_{train} = 0.0$ | $66.21 \pm 0.30$ | $62.89 \pm 0.65 \pm 0.37$ | $54.26 \pm 3.07 \pm 1.30$ | $41.59 \pm 1.90 \pm 2.23$ | $36.99 \pm 2.64 \pm 2.06$ |
| $\sigma_{train} = 0.02$ | $65.76 \pm 0.34$ | $63.27 \pm 0.29 \pm 0.15$ | $59.67 \pm 0.52 \pm 0.19$ | $53.54 \pm 1.21 \pm 0.31$ | $43.65 \pm 2.32 \pm 0.65$ |
| $\sigma_{train} = 0.03$ | $65.73 \pm 0.32$ | $63.52 \pm 0.34 \pm 0.28$ | $60.14 \pm 0.68 \pm 0.42$ | $54.44 \pm 1.24 \pm 0.76$ | $45.28 \pm 2.31 \pm 1.45$ |
| $\sigma_{train} = 0.06$ | $65.90 \pm 0.10$ | $64.09 \pm 0.36 \pm 0.13$ | $61.42 \pm 0.63 \pm 0.16$ | $57.2 \pm 1.08 \pm 0.37$ | $50.68 \pm 1.91 \pm 0.82$ |
| $\sigma_{train} = 0.08$ | $65.67 \pm 0.39$ | $64.17 \pm 0.44 \pm 0.46$ | $\mathbf{62.19} \pm 0.70 \pm 0.58$ | $\mathbf{59.01} \pm 1.16 \pm 0.79$ | $\mathbf{54.26} \pm 1.95 \pm 1.24$ |
| $\rho = 0.2$ | $68.32 \pm 0.09$ | $64.5 \pm 1.03 \pm 0.56$ | $54.02 \pm 1.15 \pm 1.26$ | $49.37 \pm 0.99 \pm 0.55$ | $45.93 \pm 1.48 \pm 0.86$ |
| $\rho = 0.3$ | $68.87 \pm 0.25$ | $61.86 \pm 4.16 \pm 1.80$ | $53.15 \pm 2.30 \pm 1.60$ | $47.93 \pm 1.16 \pm 0.57$ | $44.67 \pm 1.79 \pm 0.49$ |
| $\rho = 0.4$ | $\mathbf{69.42} \pm 0.12$ | $\mathbf{64.89} \pm 2.55 \pm 3.27$ | $52.98 \pm 2.62 \pm 2.26$ | $44.41 \pm 1.03 \pm 2.10$ | $41.34 \pm 1.50 \pm 2.44$ |

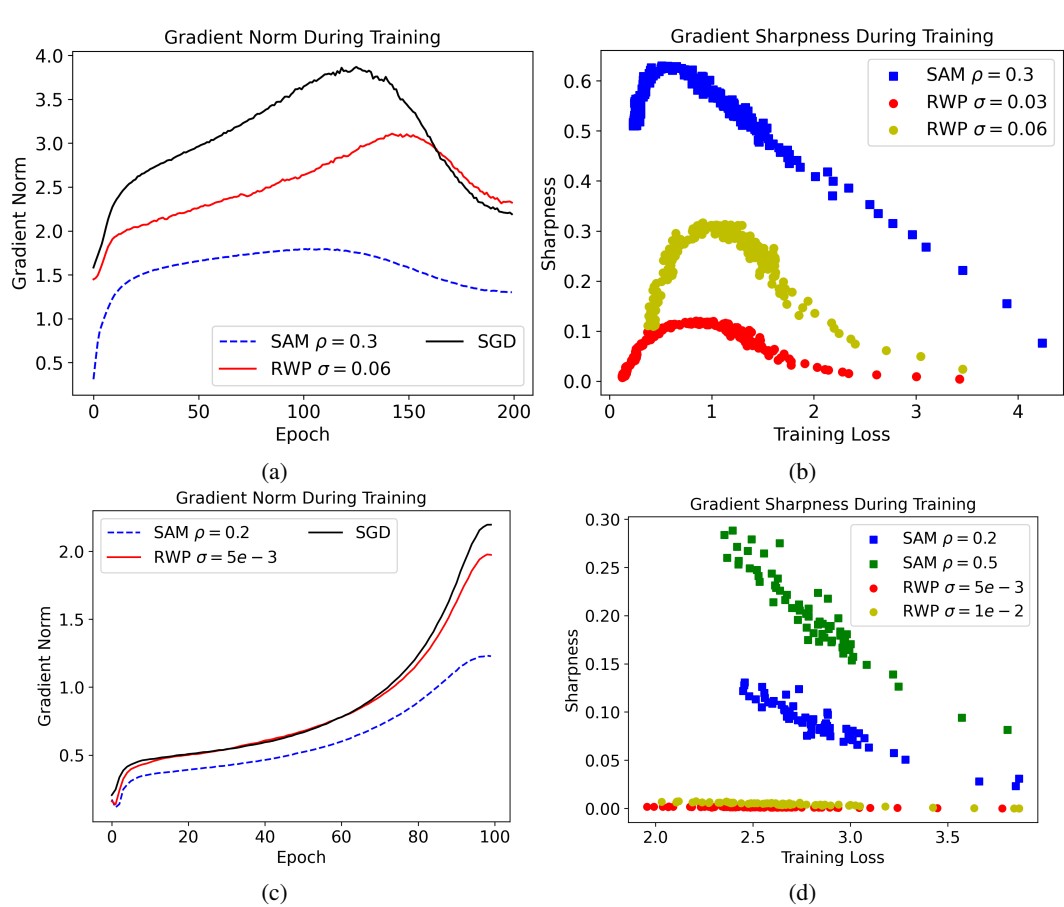

(a)

(b)

(c)

(d)

Figure 11: Plots of (a), (c) gradient norm and (b), (d) gradient sharpness for ResNet-18 during Tiny-Imagenet and ImageNet-100 training, respectively.

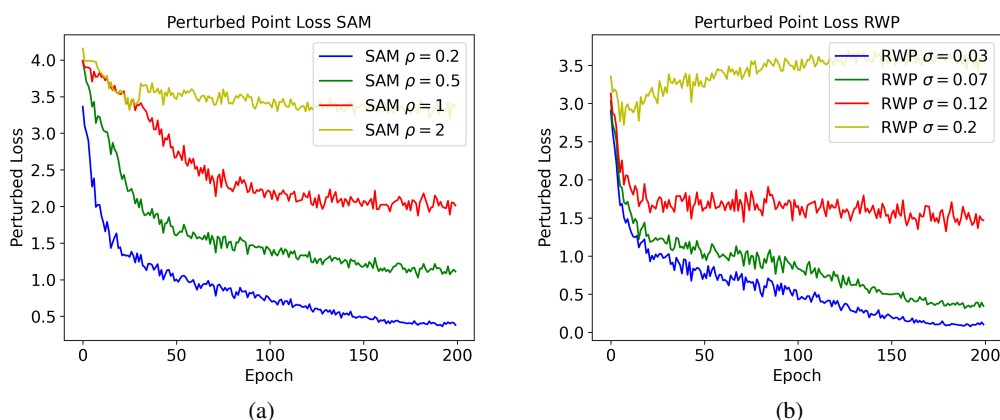

(a)                (b)

Figure 12: Plot of the perturbed loss $L(w_p)$ on Cifar-100 for (a) SAM and (b) RWP.

Table 11: Comparison of ResNet-18 models on $\sigma_{test} = 0.07$ trained using RWP for varying perturbation strength $\sigma_{train}$ when a quadratic ramp schedule is applied.

|  | $iter = 35000$ | $iter = 45000$ | $iter = 55000$ |
|---|---|---|---|
| $\sigma_{max} = 0.08$ | $61.74 \pm 3.08 \pm 0.52$ | $63.32 \pm 2.25 \pm 0.74$ | $62.55 \pm 2.71 \pm 1.61$ |
| $\sigma_{max} = 0.1$ | $63.03 \pm 2.41 \pm 0.93$ | $\mathbf{65.57} \pm 2.05 \pm 0.64$ | $64.93 \pm 1.96 \pm 1.00$ |
| $\sigma_{max} = 0.12$ | $59.37 \pm 2.16 \pm 1.89$ | $62.50 \pm 1.87 \pm 0.96$ | $63.41 \pm 2.25 \pm 1.78$ |

high loss plateau is increasing. Based on this logic, we conclude that the critical perturbation length does in fact increase during training.

## F.2 ABLATIONS

To determine the optimal schedule hyperparameters (namely the maximal perturbation strength and number of warm-up iterations), we perform a grid search, shown in Tab. 11 and Tab. 12 for RWP and SAM, respectively.

## F.3 VISUALIZING RAMPED PERTURBATION SHARPNESS

To understand the effect perturbation schedules have on sharpness evolution during training, we plot both gradient sharpness and average-direction sharpness of the quadratic schedule SAM and RWP, comparing against the constant-perturbation versions (Fig. 13). Here, we observe a coherent trend: in the early stages of training, the use of the ramp schedules reduces gradient sharpness significantly, after which the growing perturbations cause the gradient sharpness to surge. In the case of average-direction sharpness, shown in Fig. 13b, we see that while training using the schedules increases the training loss at convergence, sharpness is reduced compared to the constant perturbation baselines. This is especially noticeable in the case of SAM: we recall from Fig. **??** that increasing constant-perturbation $\rho$ from 0.2 to 0.5 *has no effect on reducing average-direction sharpness*. However, when using the quadratic perturbation schedule, a drastic reduction in sharpness can now be achieved through large perturbations.

Table 12: Comparison of ResNet-18 models on $\sigma_{test} = 0.07$ trained using SAM for varying perturbation strength $\rho$ when a quadratic ramp schedule is applied.

|  | $iter = 15000$ | $iter = 20000$ | $iter = 25000$ |
|---|---|---|---|
| $\rho_{max} = 0.8$ | $57.64 \pm 3.01 \pm 1.80$ | $59.71 \pm 1.93 \pm 1.66$ | $58.35 \pm 3.81 \pm 0.74$ |
| $\rho_{max} = 1.0$ | $57.54 \pm 2.34 \pm 0.52$ | $\mathbf{60.24} \pm 1.98 \pm 0.83$ | $59.13 \pm 2.40 \pm 0.68$ |
| $\rho_{max} = 1.2$ | $57.07 \pm 1.92 \pm 0.33$ | $58.83 \pm 1.78 \pm 1.14$ | $56.97 \pm 1.92 \pm 1.87$ |

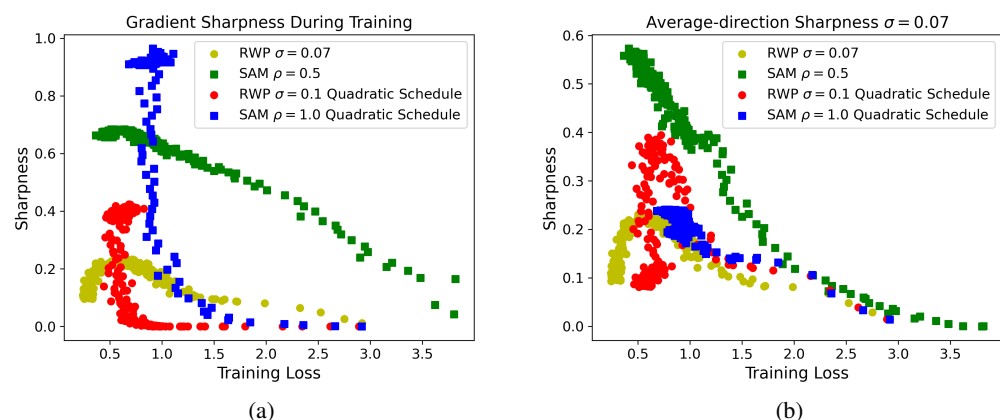

(a)                                                                      (b)

Figure 13: Plots of (a) gradient sharpness and (b) ascent-direction sharpness as a function of Cifar-100 training loss for a ResNet-18 trained using quadratic schedule SAM and RWP.

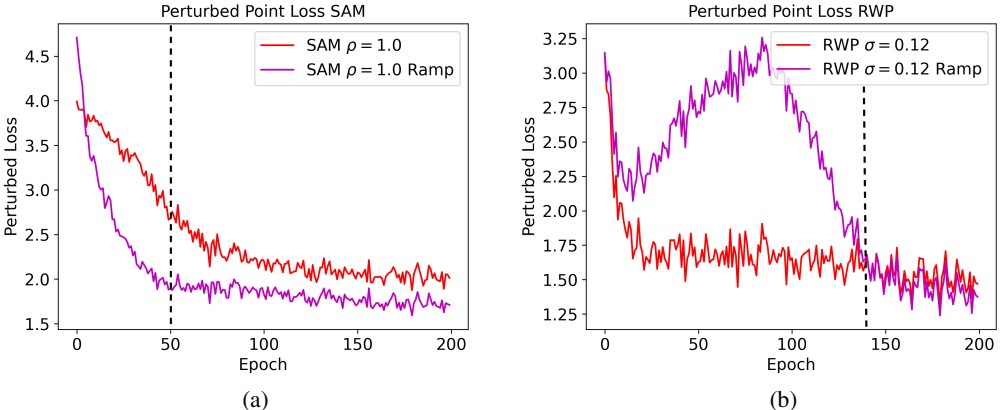

(a)                                                                      (b)

Figure 14: Comparison of the perturbed loss $L(w_p)$ for ramped/constant schedules of (a) SAM and (b) RWP on Cifar-100. The perturbed loss is calculated at a constant perturbation distance (equal to the maximum perturbation), including for the dynamic schedules. Dotted black line denotes epoch at which ramped perturbation strength reaches strength equal to constant perturbation.

### F.4 ALTERNATIVE VIEW OF LANDSCAPE BROADENING

To provide another view of how the dynamic perturbation schedule affects the training dynamics, we again plot the perturbed training loss as a function of epoch comparing the quadratically-ramped/constant schedules with equivalent terminal perturbation strengths for both RWP and SAM, shown in Fig. 14. In these plots, we observe that for the dynamic perturbation schedules, even after the initial warm-up period (meaning the size of the perturbation has reached the terminal value), the perturbed training loss is *lower* than that of the constant schedule. This difference is particularly stark for SAM: at the final epoch of the ramp, there is $\tilde{0}.75$ difference in perturbed loss between the two trajectories. This clearly illustrates the contrast in evolution of the two trajectories: on top of the natural decline of the critical perturbation length, the dynamic schedules guide the optimization towards a flatter basin than would otherwise be encountered, further reducing the critical perturbation length.

