# OpenReview forum: "Incorruptible Neural Networks: Training Models that can Generalize to Large Internal Perturbations"
_ICLR.cc/2026/Conference — ICLR 2026 Conference Withdrawn Submission_

### Official Review · Reviewer_1dpx · 2025-10-28

**Soundness:** 3
**Presentation:** 3
**Contribution:** 2
**Rating:** 6
**Confidence:** 3

**Summary:**

This paper investigates the model's robustness to weight perturbations, which may appear on real-world hardware devices. Building on existing SAM and RWP studies, this work examines the weight loss landscape during the test phase and observes that over-regularization is key for achieving optimal robustness. The paper further proposes a dynamic perturbation strategy to mitigate the negative impact of overly large perturbations on model performance.

**Strengths:**

1. It investigates the model's robustness to weight perturbations, which corresponds to the flatness of the weight loss landscape during the test phase, distinguishing it from previous research on SAM and RWP.
2. The paper is clearly written, the figures and tables are easy to understand, and the overall flow of the text is good.
3. The experiments in this paper are comprehensive. The paper conducts a systematic evaluation of SAM and RWP under various noise settings, and its proposed dynamic perturbation schedule is validated across multiple experimental setups, demonstrating that it can produce minima with significantly higher noise robustness than other methods.

**Weaknesses:**

1. The motivation is ambiguous. This paper associates the model's robustness to weight perturbations with AIMC hardware errors, which is a relatively novel scenario. However, it is not clear that the methods and experiments in this paper can be reliably transferred to real AIMC hardware.
2. The practical value of this article is unclear. The paper does not clearly demonstrate the effectiveness of its proposed method in real-world hardware deployment scenarios, such as evaluation using a target hardware's specific dataset or error profile. This leaves the practical application value of the work ambiguous.

**Questions:**

1. Could the authors provide more sufficient experiments to validate the phenomena and methods presented in this paper, such as broader noise modeling (hardware-specific noise modeling) or more complex network architectures (lightweight designs)?
2. Are there more application scenarios for examining the weight loss landscape during the test phase, especially those related to deep learning and neural network tasks? Relevant discussion would make the contribution of this paper more explicit.

---

> ### Author Response · Authors · 2025-11-21
>
> We thank you for your positive assessment of our work! Please see our responses to your questions below:
>
> **W1**
> > The motivation is ambiguous. This paper associates the model's robustness to weight perturbations with AIMC hardware errors, which is a relatively novel scenario. However, it is not clear that the methods and experiments in this paper can be reliably transferred to real AIMC hardware.
>
> We have clarified our motivation with an explanation of why our noise setting is a valid model of AIMC hardware errors (please see top-level comment).
>
> **Q1**
> > Could the authors provide more sufficient experiments to validate the phenomena and methods presented in this paper, such as broader noise modeling (hardware-specific noise modeling) or more complex network architectures (lightweight designs)?
>
> We have added new experiments in which we simulate inference on multiple analog memory devices, namely RRAM [1] and SONOS [2]. These results show favorable performance of our trained models under real analog noise settings, demonstrating that our conclusions hold for a practical setting of interest. See Section 6 of the revised draft.
>
> **Q2**
> >Are there more application scenarios for examining the weight loss landscape during the test phase, especially those related to deep learning and neural network tasks? Relevant discussion would make the contribution of this paper more explicit.
>
> We do believe the problem of finding flat test loss minima is of relevance to problems beyond hardware noise. For example, defense against adversarial attack is a similar scenario in which internal weight robustness is of interest. However, we have not experimentally investigated the application of our methods to this setting, and as such we limit our discussion on this topic.
>
> [1] Milo et. al. IEEE International Reliability Physics Symposium 2021 "Optimized programming algorithms for multilevel rram in hardware neural networks."
>
> [2] Xiao et. al. IEEE Transactions on Circuits and Systems 2022 "An accurate, error-tolerant, and energy-efficient neural network inference engine based on sonos analog memory."

---

> ### Comment · Reviewer_1dpx · 2025-11-26
>
> I sincerely thank the authors for their explanations and the additional experiments. The rebuttal has addressed some of my concerns. However, after considering the comments from other reviewers and the authors’ responses, I have decided to lower my confidence score from 3 to 1.
>
> Additionally, I would like to offer two small suggestions that may help with future revisions: (1) in the current manuscript, in-text citations are sometimes difficult to distinguish from the main text, so clearer formatting would be helpful; and (2) in the revised version, highlighting the updated parts would make it easier for reviewers to track the changes.

---

### Official Review · Reviewer_36fX · 2025-10-28

**Soundness:** 3
**Presentation:** 2
**Contribution:** 2
**Rating:** 2
**Confidence:** 3

**Summary:**

This work empirically studies SAM and RWP against weight perturbations, and design a  dynamic scheme for enhancement.

**Strengths:**

It presents in-depth comparisons between SAM and RWP, providing more extensive evaluations on top of the existing findings.

**Weaknesses:**

1. This work seems an empirical study, where no theoretical analyses or analytical discussions are provided, making limited contributions to the fundamentals in this topic.

2. As being positioned from a rather empirical perspective, it would be more solid to expand the evaluations, i.e., Transformer-based architectures, tasks/datasets beyond imaging processing, and so on.

3. By intuitive speculations, results from sec 4.2 and 4.3 are not difficult to conceptualize. Despite the observations presented in this work, it is still taking efforts to determine the perturbation strength (we still need a lot tuning in the experiments). Hence, from the reviewer's perspective and understanding, it is a bit obscure to truly seize the fundamental and key contribution here.

4. For the dynamic schedule, it is rather heuristic and remains with efforts to do quite some tuning, which hinders its contributions practical-wisely.

5. In general, this work is a bit tedious in writing and wordy, where all analyses, investigations, results explanations are all by text. It would be good with some analytical discussions and other way of presenting. The reviewer recommend the authors to clarify the novelty, and its soundness and practical impact to the field with more in-depth analyses, better with enhance evaluations under more varied settings in model structures, datasets, and tasks, rather than the current plain comparisons between SAM and RWP.

**Questions:**

See weakness.

---

> ### Author Response · Authors · 2025-11-21
>
> We thank you for your comprehensive review, and offer our responses to your concerns below:
>
> **W1**
> > This work seems an empirical study, where no theoretical analyses or analytical discussions are provided, making limited contributions to the fundamentals in this topic.
>
> We acknowledge that our work is empirical in nature. However, we believe our expanded experiments in addition to the practical relevance of our paper to hardware deployment provide a significant degree of insight to a relatively novel and under-explored problem setting.
>
> **W2**
> > As being positioned from a rather empirical perspective, it would be more solid to expand the evaluations, i.e., Transformer-based architectures, tasks/datasets beyond imaging processing, and so on.
>
> We appreciate the need for an expanded experimental validation. As mentioned in the top comment, we have performed a plethora of new experiments with more backbones and datasets, which we present in Appendix Sec D and F.
>
> **W3**
> > By intuitive speculations, results from sec 4.2 and 4.3 are not difficult to conceptualize. Despite the observations presented in this work, it is still taking efforts to determine the perturbation strength (we still need a lot tuning in the experiments). Hence, from the reviewer's perspective and understanding, it is a bit obscure to truly seize the fundamental and key contribution here.
>
> We agree that our approach to noise-robustness still requires a significant degree of manual tuning. However, we believe demonstrating that optimal performance does not arise from matching training time to test time perturbations is a significant result in and of itself, as this challenges previously-held assumptions on training noise-robust neural networks. In addition, our own improvements to SAM and RWP training, in combination with our detailed analysis on their training dynamics, will provide a solid foundation for future exploration into this direction.
>
> **W4**
> >For the dynamic schedule, it is rather heuristic and remains with efforts to do quite some tuning, which hinders its contributions practical-wisely.
>
> We agree that this approach is heuristic, but again argue that the insight this result provides into the dynamics of perturbative training (namely that large perturbations are beneficial, but cannot be tolerated in early epochs) is of practical importance to designing future work into noise-aware training.
>
> **W5**
> >In general, this work is a bit tedious in writing and wordy, where all analyses, investigations, results explanations are all by text. It would be good with some analytical discussions and other way of presenting. The reviewer recommend the authors to clarify the novelty, and its soundness and practical impact to the field with more in-depth analyses, better with enhance evaluations under more varied settings in model structures, datasets, and tasks, rather than the current plain comparisons between SAM and RWP.
>
> We appreciate the reviewer's assessment of our writing, and hope our strengthened introduction and enhanced experiments improve the soundness of the paper.

---

### Official Review · Reviewer_iybT · 2025-10-28

**Soundness:** 1
**Presentation:** 2
**Contribution:** 1
**Rating:** 2
**Confidence:** 3

**Summary:**

This paper investigates the weight robustness of neural networks trained using Sharpness-Aware Minimization (SAM) and Random Weight Perturbation (RWP). The authors find that using stronger perturbations during training than those at test time leads to optimally noise-robust models. They also identify a vanishing-gradient effect in SAM when strong perturbations are applied. To address this, they introduce a dynamic perturbation schedule that gradually increases perturbation magnitude throughout training, aligning with the evolving loss landscape and resulting in models with significantly improved robustness to weight noise.

**Strengths:**

The demonstrations in Figures 3 and 4 are clear and informative. They effectively illustrate how the loss, gradient norm, and sharpness evolve during training, revealing how SAM and RWP interact with the geometry of the loss surface. The experiments are well-controlled, with systematic variations in hyperparameters to isolate specific effects. These analyses lead to meaningful conclusions about the vanishing-gradient phenomenon in SAM and the differing robustness characteristics of SAM and RWP.

**Weaknesses:**

1. The motivation for this work is not clearly articulated. As stated on page 1, paragraph 2, the authors refer to analog in-memory computing (AIMC) and “hardware errors” as the practical motivation. However, the discussion remains vague. The paper briefly claims that prior works lack “a broad understanding of noise-robustness in neural networks” and “a connection of these efforts to existing flatness-finding approaches such as SAM and RWP,” but these statements are overly general and do not convincingly establish the importance or novelty of addressing these issues.
Moreover, regarding “hardware errors,” it is unclear what specific form these errors take. Are they modeled as random Gaussian perturbations, or do they follow structured patterns such as quantization errors, conductance drift, or loss of precision? Since the type and distribution of hardware noise fundamentally determine the robustness objective, this assumption is crucial and should be explicitly defined and justified. Without a clear characterization of the error model, the practical relevance of the results to real AIMC systems remains uncertain.

2. On page 4, the authors state that all experiments are conducted using ResNet-18 on the CIFAR-100 dataset. While this setup is reasonable for preliminary investigation, it represents a relatively small-scale benchmark. Consequently, the conclusions about over-regularization, vanishing gradients, and noise robustness may not generalize to larger or more complex architectures and datasets.

3. The paper lacks theoretical justification or analytical evidence supporting the observed phenomena. The findings are entirely empirical, relying on trends specific to this experimental configuration. To substantiate the general claims about the relationship between perturbation strength, gradient behavior, and robustness, additional experiments on larger-scale datasets (e.g., ImageNet) or complementary theoretical analysis would be necessary. Without such validation, the conclusions remain suggestive rather than definitive.

**Questions:**

1. Theoretical background:
While the paper includes a Preliminaries section, it does not provide a complete theoretical framework or formal theorems supported by clear assumptions. I would expect at least a subsection devoted to theoretical analysis that establishes or justifies the observed phenomena, such as the relationship between perturbation strength, flatness, and gradient behavior. I suggest that future revisions include a more rigorous theoretical component to substantiate the empirical findings.

2. Experimental coverage:
In the absence of sufficient theoretical support, the experimental evaluation should be broadened to ensure the robustness and generality of the conclusions. Specifically, experiments should include a wider range of architectures, datasets, or even different tasks to demonstrate that the proposed methods and observations hold beyond the ResNet-18 and CIFAR-100 setting.

---

> ### Author Response · Authors · 2025-11-21
>
> We thank you for the detailed review. We hope our responses below address your primary concerns:
>
> **W1**
> > The motivation for this work is not clearly articulated. As stated on page 1, paragraph 2, the authors refer to analog in-memory computing (AIMC) and “hardware errors” as the practical motivation. However, the discussion remains vague. The paper briefly claims that prior works lack “a broad understanding of noise-robustness in neural networks” and “a connection of these efforts to existing flatness-finding approaches such as SAM and RWP,” but these statements are overly general and do not convincingly establish the importance or novelty of addressing these issues.
>
> As stated in the top comment, we have rewritten the introduction to better articulate our motivation. Prior works have relied on the foundational assumption that resilience to analog errors is best addressed through inducing the same expected error distribution during training (an intuitive, but ultimately unsupported idea). Through our detailed experimentation, our work counters this previously-held notion.
>
> > Moreover, regarding “hardware errors,” it is unclear what specific form these errors take. Are they modeled as random Gaussian perturbations, or do they follow structured patterns such as quantization errors, conductance drift, or loss of precision? Since the type and distribution of hardware noise fundamentally determine the robustness objective, this assumption is crucial and should be explicitly defined and justified. Without a clear characterization of the error model, the practical relevance of the results to real AIMC systems remains uncertain.
>
> As mentioned in the top comment, we argue that zero-mean Gaussian noise is a valid model of analog programming errors, based on the central limit theorem argument. We further empirically support this assumption with our simulations of analog hardware inference, demonstrating that our models show improved noise-robustness on simulated hardware. Although quantization errors and drift are of concern to analog systems, we restrict our analysis to the programming error component in this work.
>
> **Q1**
> > Theoretical background: While the paper includes a Preliminaries section, it does not provide a complete theoretical framework or formal theorems supported by clear assumptions. I would expect at least a subsection devoted to theoretical analysis that establishes or justifies the observed phenomena, such as the relationship between perturbation strength, flatness, and gradient behavior. I suggest that future revisions include a more rigorous theoretical component to substantiate the empirical findings.
>
> We acknowledge that our work is empirical in nature. However, we believe our expanded experiments in addition to the practical relevance of our paper to hardware deployment provide a significant degree of insight to a relatively novel and under-explored problem setting. We hope that you will re-consider in light of our increased experimental evidence.
>
> **Q2**
> > Experimental coverage: In the absence of sufficient theoretical support, the experimental evaluation should be broadened to ensure the robustness and generality of the conclusions. Specifically, experiments should include a wider range of architectures, datasets, or even different tasks to demonstrate that the proposed methods and observations hold beyond the ResNet-18 and CIFAR-100 setting.
>
> We appreciate the need for an expanded experimental validation. As mentioned in the top comment, we have performed a plethora of new experiments with more backbones and datasets, which we present in Appendix Sec D and F.

---

### Official Review · Reviewer_iu3d · 2025-10-29

**Soundness:** 3
**Presentation:** 2
**Contribution:** 3
**Rating:** 4
**Confidence:** 4

**Summary:**

This paper investigates model robustness in the weight space by introducing two loss functions, SAM and RWP. It yields two key findings: (1) over-regularized training with strong perturbations yields the most robust weights; (2) strong perturbations in the weight space could induce a vanishing gradient effect due to increased sharpness in the loss landscape.

**Strengths:**

**S1.** Investigating the impact of weight perturbations at test time is of significant interest, as it represents a practical challenge in real-world systems affected by hardware noise.

**S2.** The authors conduct extensive experiments analyzing the loss landscape, sharpness, backbones, and the effects of varying $\sigma$ values.

**Weaknesses:**

**W1.** Since model parameters are often not externally accessible, I think the premise of manually perturbing weights has limited practical relevance. However, I agree with the argument in the introduction that hardware can inject noise into model weights. Therefore, investigating robustness against real hardware noise patterns is crucial. In this work, the perturbations are distributed as zero-mean isotropic Gaussian. How this assumption aligns with real-world noise patterns?

**W2.** As shown in Tables 1 and 2, model robustness varies significantly with different configurations of $\sigma_{test}$, $\sigma_{train}$, and $\rho$. This implies that a model must be retrained specifically for different test-time noise profiles (i.e., different hardware), which is computationally expensive. A more critical issue arises when the hardware noise characteristics are unknown, making it impractical to determine the appropriate training parameters.

**W3.** The experimental validation is limited to relatively shallow networks (e.g., ResNet-18, WRN-16). It remains unclear whether the observed benefits of SAM and RWP would scale to deeper architectures, which is critical for assessing the general applicability of the proposed methods.

**W4.** The practical impact of this work would be significantly strengthened by validation on real hardware. The current approach, which relies on simplifying Gaussian assumptions to simulate noise, may not fully capture the complex characteristics of actual analog systems, thereby limiting the persuasiveness of the findings.

**Questions:**

Please address **W1-W3** in the weakness section.

---

> ### Author Response · Authors · 2025-11-21
>
> We thank you for the detailed review and questions. We offer our responses to your questions below:
>
> **W1**
> > Since model parameters are often not externally accessible, I think the premise of manually perturbing weights has limited practical relevance.
>
>  We acknowledge that in certain deployment scenarios model parameters cannot be externally accessed. Nonetheless, models which are trained/fine-tuned with the goal of deployment in a specific hardware setting can benefit from a perturbative training approach.
> > Therefore, investigating robustness against real hardware noise patterns is crucial. In this work, the perturbations are distributed as zero-mean isotropic Gaussian. How this assumption aligns with real-world noise patterns?
>
> To demonstrate the relevance of our results to real hardware, we perform simulated inference experiments using hardware noise patterns from two real memory devices, RRAM [1] and SONOS [2] (see Section 6 of the revised draft). In summary, our models demonstrate enhanced performance over the baseline in the realistic hardware settings. Please see the top-level comment for the explanation on the validity of the assumption of zero-mean Gaussian noise.
>
> **W2**
> >As shown in Tables 1 and 2, model robustness varies significantly with different configurations of , , and . This implies that a model must be retrained specifically for different test-time noise profiles (i.e., different hardware), which is computationally expensive. A more critical issue arises when the hardware noise characteristics are unknown, making it impractical to determine the appropriate training parameters.
>
> We agree that this is a shortcoming of our current understanding of noise-aware training. However, we believe that our work lays foundational groundwork for understanding general properties of noise-robustness, which we hope to explore in more depth in future works.
>
> **W3**
> > The experimental validation is limited to relatively shallow networks (e.g., ResNet-18, WRN-16). It remains unclear whether the observed benefits of SAM and RWP would scale to deeper architectures, which is critical for assessing the general applicability of the proposed methods.
>
> We have greatly expanded our experimental validation with deeper backbone networks (e.g. ResNet-50) in Appendix Sec D. Our results show that SAM and RWP do continue to show improved performance for these larger backbone models.
>
> **W4**
> > The practical impact of this work would be significantly strengthened by validation on real hardware. The current approach, which relies on simplifying Gaussian assumptions to simulate noise, may not fully capture the complex characteristics of actual analog systems, thereby limiting the persuasiveness of the findings.
>
> Due to the limitations of memory cell size in currently-available hardware, running neural network inference on the scale of modern backbone models is unfortunately not feasible. We hope that our simulated hardware results convincingly support our findings.
>
> [1] Milo et. al. IEEE International Reliability Physics Symposium 2021 "Optimized programming algorithms for multilevel rram in hardware neural networks."
>
> [2] Xiao et. al. IEEE Transactions on Circuits and Systems 2022 "An accurate, error-tolerant, and energy-efficient neural network inference engine based on sonos analog memory."

---

> > ### Comment · Reviewer_iu3d · 2025-11-24
> > **Comment After the Rebuttal**
> >
> > Thank the authors for the response. I appreciate the additional explanations and experiments, which indeed help clarify the work and enhance the paper’s contribution to some extent. However, I believe that such substantial revisions to the introduction and main experiments are not appropriate at this stage (the main text is 10-page now). Moreover, even in the revised version, further clarification is still needed regarding the paper’s limitations and its actual contributions. Therefore, I decide to keep my score.
> >
> > By the way, for future rebuttal stages, I would suggest that the authors place the additional explanations and experiments requested by reviewers in the rebuttal document rather than incorporating them directly into the main text, as this makes it easier for reviewers to assess the changes.

---

### Author Response · Authors · 2025-11-21

We kindly thank the reviewers for their detailed and constructive comments. We appreciate that the reviewers recognize the importance of training neural networks robust to weight perturbations and its relation to analog hardware deployment. To strengthen our paper, we have performed extensive new experiments, while also re-writing sections of the text that were unclear. We acknowledge that several concerns were raised by multiple reviewers, and we summarize these and our response to them below:

* **Motivation is not clearly stated (reviewers iybT, 1dpx).** We have re-written the introduction section to make clear our motivation and assumptions. In summary, we are motivated to study specifically the effects of analog programming error; we argue that the modeling of these errors as zero-mean Gaussians is valid, as a) non-zero mean noise can be compensated for with a shift in the programmed conductance, and b) non-Gaussian distributions, when summed, will be Gaussian by the Central Limit Theorem. Prior works that have investigated this problem adopt the principle (without extensive validation) that the ideal training regimen should faithfully emulate the expected noise characteristics of inference. We convincingly refute this idea by showing that a) unmatched training and test noise distributions produce more robust models and b) SAM's adversarial perturbation is a more optimal training approach for certain test noise distributions.

* **The results are not validated using hardware-specific noise modeling (reviewers iu3d, iybT, 1dpx).** To demonstrate that our results generalize to real analog hardware deployment scenarios, we perform experiments using CrossSim [2], an open-source library capable of simulating neural network inference on analog hardware, and demonstrate that on multiple memory devices, our models exhibit an increased noise-robustness over the baseline. Furthermore, we find that training with our generic perturbation distributions outperforms training with the exact device error profile, adding further emphasis to our statement in the previous bullet point. See Section 6 in the revised draft.

* **Experimental results are limited, focusing solely on ResNet/Cifar-100 classification scenario (reviewers iu3d, iybT, 36fx).** To validate that our results generalize and are not specific to a narrow experimental setting, we vastly expand our experimental results, investigating multiple backbones (e.g. ResNet-50, Pyramidnet-50) and datasets (e.g. ImageNet-100, Tiny-ImageNet). We show that the same phenomena observed in our initial experiments appear across these diverse settings, confirming the generality of our results. See Appendix Section D and E for more details.

In addition, we have provided detailed responses to each of the reviewer's questions individually.

[1] Rasch et. al. Nature Comms. 2023 "Hardware-aware training for large-scale and diverse deep learning inference
workloads using in-memory computing-based accelerators."

[2] Feinberg et. al. "Crosssim: accuracy simulation of analog in-memory computing"

---

### Note · Authors · 2026-01-19

I have read and agree with the venue's withdrawal policy on behalf of myself and my co-authors.